# OPENCHAT: ADVANCING OPEN-SOURCE LANGUAGE MODELS WITH MIXED-QUALITY DATA

**Guan Wang**[1,2]*, **Sijie Cheng**[1,3,5]*, **Xianyuan Zhan**[3,4], **Xiangang Li**[5], **Sen Song**[2] ✉, **Yang Liu**[1,3,4] ✉

[1]Department of Computer Science and Technology, Tsinghua University
[2]Laboratory of Brain and Intelligence, Tsinghua University
[3]Institute for AI Industry Research (AIR), Tsinghua University
[4]Shanghai Artificial Intelligence Laboratory [5]01.AI
imonenext@gmail.com, csj23@mails.tsinghua.edu.cn

## ABSTRACT

Nowadays, open-source large language models like LLaMA have emerged. Recent developments have incorporated supervised fine-tuning (SFT) and reinforcement learning fine-tuning (RLFT) to align these models with human goals. However, SFT methods treat all training data with mixed quality equally, while RLFT methods require high-quality pairwise or ranking-based preference data. In this study, we present a novel framework, named OpenChat, to advance open-source language models with mixed-quality data. Specifically, we consider the general SFT training data, consisting of a small amount of expert data mixed with a large proportion of sub-optimal data, without any preference labels. We propose the C(onditioned)-RLFT, which regards different data sources as coarse-grained reward labels and learns a class-conditioned policy to leverage complementary data quality information. Interestingly, the optimal policy in C-RLFT can be easily solved through single-stage, RL-free supervised learning, which is lightweight and avoids costly human preference labeling. Through extensive experiments on three standard benchmarks, our openchat-13b fine-tuned with C-RLFT achieves the highest average performance among all 13b open-source language models. Moreover, we use AGIEval to validate the model generalization performance, in which only openchat-13b surpasses the base model. Finally, we conduct a series of analyses to shed light on the effectiveness and robustness of OpenChat. Our code, data, and models are publicly available at https://github.com/imoneoi/openchat and https://huggingface.co/openchat.

## 1 INTRODUCTION

Recently, there have been notable advancements in Large Language Models (LLMs), such as GPT-4 (OpenAI, 2023) and Chinchilla (Hoffmann et al., 2022), demonstrating impressive performance in various downstream natural language processing (NLP) tasks (Zhao et al., 2023). Despite the remarkable success of GPT-4, the specific techniques employed in its development remain shrouded in mystery. To gain a deeper understanding of the underlying technical aspects and to promote the widespread adoption of LLMs, a series of open-source base language models have emerged, especially LLaMA (Touvron et al., 2023a) and Llama 2 (Touvron et al., 2023b). Building upon the released base language models, there are typically two methods to align these base models to specific abilities, including supervised fine-tuning (SFT) and reinforcement learning fine-tuning (RLFT).

The first line of methods (Chiang et al., 2023; Taori et al., 2023) use SFT to enhance instruction following abilities. Most existing methods primarily focus on designing SFT datasets. Some studies (Chiang et al., 2023; Geng et al., 2023) collect user-shared conversations as well as human feedback datasets from the public web, while others (Xu et al., 2023a; Ding et al., 2023) develop frameworks for automatically gathering extensive open-domain instructions spanning various difficulty levels. However, these constructed SFT datasets are generally mixed with limited expert data

---

*Equal contribution

Figure 1: Our proposed framework OpenChat with Conditioned-RLFT to advance the open-source language model fine-tuning with mixed-quality data, comparing to previous supervised fine-tuning (SFT) method and reinforcement learning fine-tuning (RLFT) method. MLE and RL denote maximum likelihood estimates and reinforcement learning, respectively.

and a large proportion of sub-optimal data due to the high cost of human labor and API requests. Naturally, it is not advisable to indiscriminately feed all these mixed conversations to the base model, as the low-quality data are likely to negatively impact learning (Zhou et al., 2023; Xu et al., 2022). However, this is largely neglected in previous methods, which often treat all training data equally.

To allow LLMs to go beyond modeling the training data distribution, recent LLMs (OpenAI, 2023; Touvron et al., 2023b) adopt RLFT to align better with the human desired behaviors, especially API-based models. The well-known reinforcement learning from human feedback (RLHF) method (Ouyang et al., 2022; Christiano et al., 2017; Bai et al., 2022b) first collects plenty of high-quality preference feedback from human annotators to fit one or multiple reward models (typically also trained based on smaller LLMs), and then uses RL to maximize the estimated reward. The involvement of learning and optimizing with extra reward models using RL brings considerable computational and stability issues. Some recent studies (Rafailov et al., 2023; Yuan et al., 2023) partly address this problem by avoiding fitting the reward model and fusing preference modeling and LLM fine-tuning into a single-stage training process. However, all existing RLHF methods require high-quality pairwise or ranking-based preference data for preference modeling, which inevitably require expensive human expert annotations (Casper et al., 2023).

To address the aforementioned limitations, we propose a new framework, named OpenChat, to advance the open-source language model fine-tuning with *mixed-quality data* as shown in Fig. 1. Here, we consider the general non-pairwise (nor ranking-based) SFT training data, consisting of a small amount of expert data and a large proportion of easily accessible sub-optimal data, without any preference labels. Specifically, we propose the Conditioned-RLFT (C-RLFT), which enables leveraging mixed-quality training data with very coarse-grained reward labels. The reward label can be as simple as a relative value differentiating different classes of data, i.e., expert and sub-optimal. We derive C-RLFT based on the KL-regularized RL framework (Jaques et al., 2019; Korbak et al., 2022), which maximizes the reward while penalizing the difference between the fine-tuned policy and a reference policy. However, to remedy the imperfect reward signal, we learn the fine-tuned LLM itself as a class-conditioned policy (i.e., conditioning data source classes with distinct prompt tokens), and regularize it with a better and more informative class-conditioned reference policy instead of the original pre-trained LLM. The optimal policy for this RL problem can be shown as equivalent to a class-conditioned reward-weighted regression problem, which can be easily solved through single-stage supervised learning. C-RLFT provides several particularly desirable features for open-source LLM fine-tuning. First, it allows for simple and RL-free training, largely removing the complexities and instabilities in typical RLHF fine-tuning. Second, it has extremely low requirements for the quality of the reward and does not need costly human feedback collection.

Despite being simple and lightweight, our proposed OpenChat with C-RLFT achieves great performance in a series of benchmark evaluations. Specifically, we leverage the ShareGPT conversations dataset[1] following Vicuna (Chiang et al., 2023) and use `llama-2-13b` as the base model. It is worth noting that our proposed method can be applied to any mixed-quality datasets and arbitrary base language models. We conduct extensive experiments on three standard benchmarks to assess instruction following ability, including Alpaca-Eval (Li et al., 2023), MT-bench (Zheng et al., 2023) and Vicuna-bench (Chiang et al., 2023). The results demonstrate that `openchat-13b` significantly surpasses previous 13b open-source language models and can even outperform `gpt-3.5-turbo` in all three benchmarks. Furthermore, we also use AGIEval (Zhong et al., 2023) to prove the generalization, where `openchat-13b` also achieves the top-1 average accuracy among all 13b open-source language models. Finally, we design a series of ablation studies and analyses to validate the contribution of different components, and performance consistency, providing insights into the effectiveness and robustness of OpenChat.

---

[1] The ShareGPT dataset is collected from https://sharegpt.com/.

## 2 PRELIMINARIES

Given a conversation dataset $\mathcal{D} = \{(x_i, y_i)\}$, where $x_i$ indicates the instruction, $y_i$ is its corresponding response, the pre-trained language model $\pi_0(y|x)$ can be regarded as a probability distribution mapping from instructions to responses. There are two lines of research to adapt the pre-trained language model $\pi_0(y|x)$ to a fine-tuned language model $\pi_\theta(y|x)$ with desirable features, including supervised fine-tuning and reinforcement learning fine-tuning.

**Supervised Fine-tuning (SFT).** This line of methods (Xu et al., 2023a;b; Ding et al., 2023) directly uses a high-quality conversation dataset $\mathcal{D}$ to fine-tune the pre-trained language model $\pi_0(y|x)$ using supervised learning, i.e., maximum likelihood estimates (MLE):

$$J_{\text{SFT}}(\theta) = \mathbb{E}_{(x,y)\sim\mathcal{D}} \left[\log \pi_\theta(y|x)\right] \tag{1}$$

where $\pi_\theta$ is initialized from $\pi_0$. To ensure the fine-tuning performance, SFT requires the conversation dataset $\mathcal{D}$ to have very high quality, because SFT treats all training data uniformly (Zhou et al., 2023; Chen et al., 2023). However, the collection of high-quality SFT datasets can be very expensive. Most existing open-source LLMs (Chiang et al., 2023; Geng et al., 2023; Xu et al., 2023a; Ding et al., 2023) fine-tune their models using conversation datasets that likely contain a large proportion of sub-optimal data due to high costs of human labor or API requests, inevitably leading to a certain level of performance degeneration.

**Reinforcement Learning Fine-tuning (RLFT).** Another intuitive approach to align LLMs is through RL, which models rewards according to human preference feedbacks (Ouyang et al., 2022; Bai et al., 2022a; Rafailov et al., 2023) or pre-defined classifiers (Wu et al., 2023), and fine-tune LLMs to maximize the reward. The reward $r(x, y)$, either modeled explicitly or implicitly, assigns high values on desirable responses and low values on bad ones to guide the alignment of the fine-tuned LLM. A popular RL framework for fine-tuning LLMs is the KL-regularized RL (Jaques et al., 2019; Korbak et al., 2022; Rafailov et al., 2023), which adds an additional KL penalty to constrain the fine-tuned LLM $\pi_\theta(y|x)$ to stay close to the base pre-trained LLM $\pi_0(y|x)$. This has been shown beneficial to avoid distribution collapse as compared to naïvely maximize reward using RL (Korbak et al., 2022). The RL objective of this series of RLFT models can be typically formulated as follows:

$$J_{\text{RLFT}}(\theta) = \mathbb{E}_{y\sim\pi_\theta}[r(x, y)] - \beta D_{\text{KL}}\left(\pi_\theta, \pi_0\right) \tag{2}$$

In existing RLFT methods, high-quality reward signals play a crucial role in ensuring improved LLM fine-tuning performance. This, however, requires collecting considerable amounts of costly pairwise (or ranking-based) human preference feedback, which poses a major challenge in the development of many open-source language models.

## 3 OPENCHAT

In this section, we introduce the OpenChat framework, which provides a new possibility to fine-tune open-source LLMs using easily collectable and mixed-quality training data without any preference labels. More specifically, we consider the setting where we are given a pre-trained LLM $\pi_0$, a small set of high-quality/expert conversation data $\mathcal{D}_{\text{exp}}$, and a larger medium-quality or sub-optimal conversation dataset $\mathcal{D}_{\text{sub}}$, we aim to fine-tune an LLM policy $\pi_\theta$ based on $\pi_0$ using only data from $\mathcal{D}_{\text{exp}} \bigcup \mathcal{D}_{\text{sub}}$. Taking the most popular SFT dataset ShareGPT used in Vicuna (Chiang et al., 2023) as an example, the distinct data sources from GPT-4 and GPT-3.5 can be regarded as $\mathcal{D}_{\text{exp}}$ and $\mathcal{D}_{\text{sub}}$, as the overall quality of GPT-3.5 conversations generally falls short when compared to that of GPT-4 conversations (OpenAI, 2023; Li et al., 2023), where detailed comparison can be found in Sec. 5.1.

Obviously, it is not possible to derive accurate and fine-grained reward signals solely based on $\mathcal{D}_{\text{exp}}$ and $\mathcal{D}_{\text{sub}}$. However, it should be noted that the quality difference between $\mathcal{D}_{\text{exp}}$ and $\mathcal{D}_{\text{sub}}$ itself can serve as implicit or weak reward signals. To make use of this coarse-grained reward information, we provide a new insight that by regularizing $\pi_\theta$ with a better and more informative class-conditioned reference policy $\pi_c$ instead of the original base pre-trained LLM $\pi_0$, we are likely to compensate for the potential deficiencies in the rewards and achieve good fine-tuning performance. In the following, we describe the details of OpenChat and its core algorithm – C-RLFT.

## 3.1 CLASS-CONDITIONED DATASET AND REWARDS

Given the SFT conversation datasets $\mathcal{D}_{\text{exp}} \bigcup \mathcal{D}_{\text{sub}}$ with different quality levels, we can replenish them with distinct sources as class labels (e.g., $c_i \in \{\text{GPT-4}, \text{GPT-3.5}\}$) and construct a class-conditioned dataset $\mathcal{D}_c = \{(x_i, y_i, c_i)\}$. We use $\pi_c(y|x, c)$ to denote class-conditioned distribution over instructions $x$ and responses $y$ in the class-conditioned dataset $\mathcal{D}_c$, which can be perceived similarly as the behavior policy of a dataset in offline RL literature (Levine et al., 2020), with the difference that $\pi_c$ is now a class-conditioned policy.

According to the different overall quality with respect to class labels, we can naturally encode coarse-grained rewards $r_c(x, y)$ in $\mathcal{D}_c$ as follows:

$$r_c(x_i, y_i) = \begin{cases} 1, & \text{if } (x_i, y_i) \in \mathcal{D}_{\text{exp}} \text{ (e.g., } c_i = \text{GPT-4)}, \\ \alpha, & \text{if } (x_i, y_i) \in \mathcal{D}_{\text{sub}} \text{ (e.g., } c_i = \text{GPT-3.5)} \quad (\alpha < 1). \end{cases} \tag{3}$$

where we regard GPT-4 conversations as expert data $\mathcal{D}_{\text{expert}}$, and GPT-3.5 conversations as sub-optimal data $\mathcal{D}_{\text{sub}}$. Meanwhile, we set $\alpha < 1$ to guide the fine-tuned model to favor more of the high-quality responses.

## 3.2 FINE-TUNING VIA C(ONDITIONED)-RLFT

As the rewards $r_c(x, y)$ in Eq. (3) are very coarse-grained, to reliably use them in RLFT, we need to provide additional sources of information to remedy their deficiencies. Here, we introduce C-RLFT, which is inspired by the insight from the goal-conditioned supervised learning in offline RL, that by conditioning on proper information in a supervised goal/outcome-conditioned policy, it is possible to recover optimized performance (Chen et al., 2021; Emmons et al., 2021). C-RLFT contains two key ingredients: 1) fine-tuning the LLM as a class-conditioned policy $\pi_\theta(y|x, c)$, and 2) regularizing $\pi_\theta$ with respect to the class information augmented reference policy $\pi_c$ rather than the original base reference policy $\pi_0$ in the KL-regularized RL framework.

**Class-conditioned policy.** Instead of directly fine-tuning an LLM from the pre-trained model $\pi_0$ as in existing methods, we model the LLM to be fine-tuned as a class-conditioned policy $\pi_\theta(y|x, c)$. This can be easily implemented by conditioning each example from different data sources using distinct conversation templates as shown below.

> **[GPT-4 Template]**      *GPT4 User:* Question`<|end_of_turn|>`*GPT4 Assistant:*

> **[GPT-3.5 Template]** *GPT3 User:* Question`<|end_of_turn|>`*GPT3 Assistant:*

To differentiate speakers, we introduce a new `<|end_of_turn|>` special token at the end of each utterance, following Zhou et al. (2023). The `<|end_of_turn|>` token functions similarly to the EOS token for stopping generation while preventing confusion with the learned meaning of EOS during pretraining. We further discuss the effect of conversation template choice in App. A.

**Policy optimization.** To compensate for the coarse-grained reward information $r_c(x, y)$, we modify the original KL-regularized RL objective Eq. (2) and instead optimize the following problem:

$$J_{\text{C-RLFT}}(\theta) = \mathbb{E}_{y \sim \pi_\theta}[r_c(x, y)] - \beta D_{\text{KL}}(\pi_\theta, \pi_c) \tag{4}$$

The idea is to use the higher-quality and more informative class-conditioned behavior policy $\pi_c$ of $\mathcal{D}_c$ for regularization, rather than the pre-trained model $\pi_0$. We adopt this design for the following reasons. First, for most existing open-source pre-trained LLMs, their performance in many cases is still inferior to the behavior policy that generates the sub-optimal data (e.g. GPT-3.5 in ShareGPT). This means that even the sub-optimal data $\mathcal{D}_{\text{sub}}$ is likely to have higher quality than $\pi_0$. Second, $\pi_c$ contains additional data source information, which can help differentiating the quality of data.

Following prior works (Peters & Schaal, 2007; Peng et al., 2019; Korbak et al., 2022; Rafailov et al., 2023), it can be shown that the optimal solution to the above KL-regularized reward maximization objective takes the following form (see App. B for detailed derivation):

$$\pi^*(y|x, c) \propto \pi_c(y|x, c) \exp\left(\frac{1}{\beta} r_c(x, y)\right) \tag{5}$$

We can thus extract the optimized policy $\pi_\theta$ by minimizing the KL divergence between $\pi^*$ under the class-conditioned dataset $\mathcal{D}_c$ (Nair et al., 2020; Korbak et al., 2022):

$$
\begin{aligned}
\pi_\theta =& \arg\min_\theta \mathbb{E}_{(x,c)\sim\mathcal{D}_c}[D_{KL}(\pi^*(\cdot|x,c)\|\pi_\theta(\cdot|x,c))] \\
=& \arg\min_\theta \mathbb{E}_{(x,c)\sim\mathcal{D}_c}\left[\mathbb{E}_{y\sim\pi^*}[-\log\pi_\theta(y|x,c)]\right] \\
=& \arg\max_\theta \mathbb{E}_{(x,y,c)\sim\mathcal{D}_c}\left[\exp\left(\frac{1}{\beta}r_c(x,y)\right)\log\pi_\theta(y|x,c)\right]
\end{aligned}
\tag{6}
$$

The last step is obtained by plugging the closed form $\pi^*$ in Eq. (5) and using the fact that $\pi_c$ is exactly the class-conditioned behavior distribution of $\mathcal{D}_c$. This suggests that the fine-tuned policy $\pi_\theta$ can be learned through a simple reward-weighted regression objective with the class-conditioned dataset $\mathcal{D}_c$. This learning objective provides a remarkably simple scheme to fine-tune open-source LLMs. It does not require accurate reward labels, but uses conditioning to differentiate good and inferior model behaviors. Moreover, after initializing $\pi_\theta$ with $\pi_0$, we no longer need to load $\pi_0$ during training, while most RLHF methods using PPO for policy optimization (Ouyang et al., 2022; Bai et al., 2022a) still need to maintain $\pi_0$ to compute the KL penalty during fine-tuning. This enables C-RLFT to save a considerable amount of computation resources during training.

**Model inference.** During the inference phase, we assume that our C-RLFT method has learned to distinguish expert and sub-optimal data distributions. Considering that we aim to exclusively generate high-quality responses for our fine-tuned class-conditioned $\pi_\theta$, we use the same specific conversation template employed in GPT-4 conversations during the training phase as below:

**[Inference template]** *GPT4 User:* Question<|end_of_turn|>*GPT4 Assistant:*

## 4 EXPERIMENTS

### 4.1 EXPERIMENTAL SETUPS

**Mixed-quality Data.** Following Vicuna (Chiang et al., 2023), we adopt a widely-used SFT dataset, the ShareGPT dataset. The ShareGPT dataset consists of approximately 70k user-shared conversations, including around 6k expert conversations generated by GPT-4 and the remaining sub-optimal conversations from GPT-3.5. We perform experiments to assess their varying quality in Sec. 5.1.

**Benchmarks.** To evaluate the instruction-following ability, we employ the three most widely recognized benchmarks, including AlpacaEval (Li et al., 2023), MT-bench (Zheng et al., 2023) and Vicuna-bench (Chiang et al., 2023). Additionally, to verify generalization, we perform all English tasks in AGIEval (Zhong et al., 2023) using zero-shot settings, which presents a collection of human-centric standardized exams. More details of the benchmarks are listed in App. C.

**Baselines.** We evaluate the most popular API-based and open-source LLMs: (1) `gpt-4` (OpenAI, 2023) and `gpt-3.5-turbo` (Ouyang et al., 2022) are highly advanced LLMs developed by OpenAI; (2) `claude` (Bai et al., 2022a) is helpful and harmless assistants by Anthropic; (3) `llama-2-chat` (Touvron et al., 2023b) series models are the most frequently used open-source models with SFT and RLHF; (4) `wizardlm` (Xu et al., 2023a), `guanaco` (Dettmers et al., 2023), `ultralm` (Ding et al., 2023) and `vicuna` (Chiang et al., 2023) are among the well-known open-source LLMs with SFT. More details are shown in App. D.

**Automatic Evaluators.** To mitigate the cost of human annotations, we follow the official evaluators according to each benchmark. Specifically, AlpacaEval employs `alpaca_eval_gpt`, while MT-bench and Vicuna-bench use `gpt-4`. It is worth noting that these benchmarks have already computed the human agreements to ensure reliability. Additionally, we further introduce `gpt-3.5` and `claude-2` to eliminate the potential self-enhancement bias in App. E.

**Metrics.** We employ three metrics following the official implementations: (1) Win rate: This metric is employed for pairwise comparisons. Given a question and two answers generated by the tested model and the target model, the LLM evaluator needs to compare these two models. The tested model receives 1 point for a win, 0.5 points for a tie, and 0 points for a loss. (2) Score: This metric is applied for single-answer grading in MT-bench, where the LLM evaluator directly judges the

| Models | Base Models | Method | AlpacaEval | MT-bench | Vicuna-bench | Average |
|---|---|---|---|---|---|---|
| **Larger than 13b** | | | | | | |
| gpt-4 | - | SFT + RLFT | 95.3 | 82.5 | 90.0 | 89.3 |
| llama-2-chat-70b | llama-2-70b | SFT + RLFT | 92.7 | 60.0 | 87.5 | 80.1 |
| claude | - | SFT + RLFT | 88.4 | 65.0 | 76.3 | 76.6 |
| gpt-3.5-turbo | - | SFT + RLFT | 86.1 | 50.0 | 50.0 | 62.0 |
| guanaco-65b | llama-65b | SFT | 71.8 | 40.6 | 49.4 | 53.9 |
| guanaco-33b | llama-33b | SFT | 66.0 | 40.6 | 54.4 | 53.7 |
| **Equal to 13b** | | | | | | |
| vicuna-v1.1-13b | llama-13b | SFT | 70.4 | 29.4 | 45.0 | 48.3 |
| wizardlm-v1.0-13b | llama-13b | SFT | 75.3 | 33.1 | 44.4 | 50.9 |
| vicuna-v1.5-13b | llama-2-13b | SFT | 78.8 | 37.2 | 47.1 | 54.4 |
| ultralm-13b | llama-13b | SFT | 80.6 | 37.2 | 50.0 | 55.9 |
| wizardlm-v1.2-13b | llama-2-13b | SFT | 89.2 | 53.1 | 80.6 | 74.3 |
| llama-2-chat-13b | llama-2-13b | SFT + RLFT | 81.1 | 55.3 | **86.9** | 74.4 |
| **openchat-13b** | llama-2-13b | C-RLFT | **89.5** | **57.5** | 85.0 | **77.3** |

Table 1: The win-rate (%) performance of the proposed openchat-13b and other popular open-source language models. The competitors are text-davinci-003 in AlpacaEval, and gpt-3.5-turbo in both MT-bench and Vicuna-bench. The **bold** scores denote the best performance, and the underline scores indicate the second-best performance.

generated answer with a score varying from 1 to 10. (3) Accuracy: This metric is used by AGIEval for multiple-choice exam questions.

**Implementation Details.** The openchat-13b is based on the llama-2-13b (Touvron et al., 2023b). We fine-tune the model for 5 epochs on the ShareGPT dataset using the AdamW optimizer with a sequence length of 4,096 tokens and an effective batch size of 200k tokens. Given that the reward weight term in Eq. (6) $(\exp(r_c/\beta))$ remains constant within a class, we simplify the process by assigning a unit weight to $\mathcal{D}_{exp}$ and the weight of 0.1 to $\mathcal{D}_{sub}$. The AdamW optimizer's hyperparameters are set as follows: $\beta_1 = 0.9, \beta_2 = 0.95, \epsilon = 10^{-5}$, and weight decay of 0.1. We employ a cosine learning rate schedule with a maximum learning rate of $6.7 \times 10^{-5}$, which decays to 10% of the maximum value. The hyperparameters remain consistent with the base model pretraining settings following Touvron et al. (2023b). However, we scale the learning rate proportionally to the square root of the batch size, following the theoretical analysis provided by Granziol et al. (2020).

## 4.2 MAIN RESULTS

In the first set of results, we compare the win-rate (%) performance of openchat-13b and other popular LLMs on three standard benchmarks to assess the instruction-following ability. The results are presented in Table 1. Among the API-based LLMs, the win rate of gpt-4 significantly outperforms all other models, demonstrating that gpt-4 maintains obvious advantages. The open-source language model llama-2-chat-70b, which employs both SFT and RLHF, is another powerful instruction-following model which surpasses claude and gpt-3.5-turbo. However, guanaco-65b and guanaco-33b lag behind other models larger than 13b. Regarding the series of 13b models, the open-source language models based on llama-13b, including vicuna-v1.1-13b, wizardlm-v1.0-13b and ultralm-13b, achieve approximately 50% average win rates across the three benchmarks. Meanwhile, the open-source language models based on llama-2-13b generally exhibit higher average win rates. Notably, wizardlm-v1.2-13b and llama-2-chat-13b achieve average win rate scores of 74.3 and 74.4, respectively, which is close to the 76.6 score of claude. Our proposed language model openchat-13b attains the highest win rate scores in both AlpacaEval and MT-bench benchmarks, and the second-highest win rate scores in Vicuna-bench. It is worth noting that the average win rate score of openchat-13b even surpasses that of claude.

In the second set of results, we present the MT-bench scores of openchat-13b and other baseline models in Fig. 2(a). The overall trends are align closely with the win rate performance. For the API-based language models, gpt-4 continues to lead by a significant margin. Notably, all API-based language models perform better than open-source language models. Among the open-source language models, the series based on llama-2-13b generally surpasses those based on llama-13b. Meanwhile, our openchat-13b achieves the highest MT-bench score, even exceeding open-source language models with much larger parameters, such as llama-2-chat-70b.

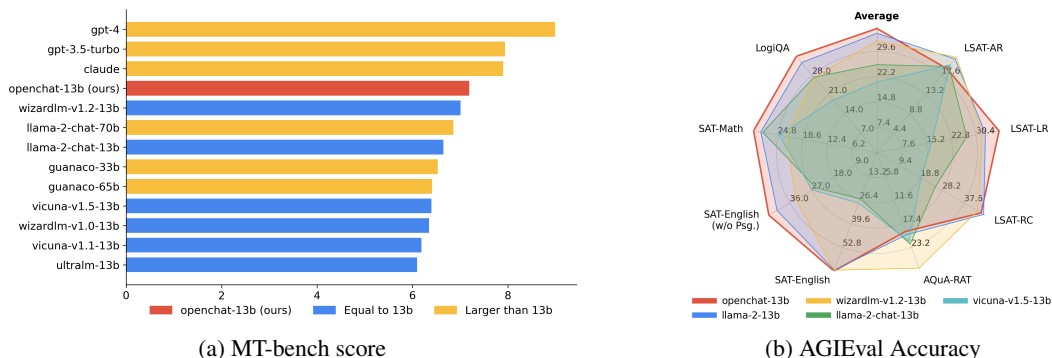

| (a) MT-bench score | (b) AGIEval Accuracy |

Figure 2: The MT-bench score (a) and AGIEval Accuracy (b) of the proposed `openchat-13b` and other popular open-source language models, where the detailed performance of AGIEval in App. F.

Finally, to further validate generalization rather than overfitting, we compare the AGIEval accuracy of the base model `llama-2-13b` and the corresponding fine-tuned models, as depicted in Fig. 2(b). It is worth noting that only the average accuracy of `openchat-13b` outperforms the base model `llama-2-13b`, while the accuracies of all other baselines have declined to varying degrees. Although `llama-2-chat-13b` excels in Vicuna-bench, the accuracy on AGIEval potentially indicates a forgetting issue.

## 5 ANALYSIS

### 5.1 DATA QUALITY DISTRIBUTION

Although prior works (OpenAI, 2023; Zhao et al., 2023) have widely confirmed that GPT-4 demonstrates superior capabilities on a broad range of tasks compared to GPT-3.5, we further analyze the quality of collected GPT-3.5 and GPT-4 conversations in the ShareGPT dataset to validate our assumption of mixed-quality data. Specifically, we randomly sample 128 conversations from each data source. Then `gpt-4` serves as the automatic evaluator, scoring the responses following Zheng et al. (2023). As illustrated in Figure 3, GPT-4 conversations contain more high-quality conversations and exhibit a higher overall score. The detailed scoring settings can be found in App. G.

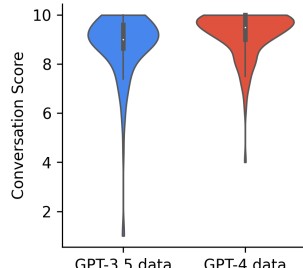

Figure 3: Quality distribution of GPT-3.5 and GPT-4 conversations in the ShareGPT dataset.

### 5.2 ABLATION STUDIES

We conduct an ablation study on the two key components of the `openchat-13b` model to ascertain their individual contributions to the overall performance, including the coarse-grained rewards and the class-conditioned policy. Additionally, we introduce two important baselines. One series is `only SFT` which uses the same ShareGPT data as OpenChat with three filtering strategies (i.e., no-filtering, only GPT-3, and only GPT-3.5). The other is `vicuna-v1.5-13b` which SFT on about 125k ShareGPT data as a baseline. The results of ablation studies are detailed in Table 2. Without coarse-grained rewards, the training phase treats different data sources equally, leading to performance decline. Similarly, without a class-conditioned policy, language models lack explicit signals to discern between expert and sub-optimal data, significantly reducing performance. We also conduct only SFT on different sources of ShareGPT data. It is worth noting that only SFT with GPT-4 data performs much better than with the entire ShareGPT data, indicating that the quality of data is much more important than quantity. However, `openchat-13b` still obviously outperforms all the `only SFT` methods, demonstrating that our proposed framework can better exploit the mixed-quality data. This indicates the significant contribution of both main components to model performance. Notably, expanding the SFT dataset from 70k to 125k has less impact on performance improvement than our proposed components, particularly the class-conditioned policy.

| | ShareGPT Dataset | AlpacaEval (text-davinci-003) | MT-bench (gpt-3.5-turbo) | Vicuna-bench (gpt-3.5-turbo) | Average |
|---|---|---|---|---|---|
| **openchat-13b** | | **89.5** | **57.5** | **85.0** | **77.3** |
| – w/o reward | ∼70k | 89.1 | 52.8 | 80.0 | 74.0 |
| – w/o condition | | 79.1 | 38.1 | 64.4 | 60.5 |
| **only SFT** | ∼70k | 78.6 | 33.1 | 46.3 | 52.7 |
| – only GPT-4 | ∼ 6k | 85.8 | 33.4 | 84.4 | 64.5 |
| – only GPT-3.5 | ∼64k | 76.5 | 16.9 | 35.0 | 42.8 |
| **vicuna-v1.5-13b** | ∼125k | 78.8 | 37.2 | 47.1 | 54.4 |

Table 2: Ablation studies of coarse-grained rewards (reward) and class-conditioned policy (condition) to `openchat-13b`.

## 5.3 REVEALING SECRETS OF C-RLFT

Firstly, we visualize the representations of `openchat-13b`, and its ablation version, `only SFT`, to distinguish between our proposed C-RLFT and SFT. We randomly sample 2,000 GPT-4 and GPT-3.5 conversations. To obtain the representations of conversations, we compute the embeddings through mean pooling of all tokens in the last Transformer layer output following Xiao (2018). These embeddings, depicted in Fig. 4, are consequently mapped to 2-D space using UMAP (McInnes et al., 2018). Multiple clusters are observed in both `only SFT` and `openchat` models, likely due to the diverse conversation domains in the ShareGPT dataset. More importantly, the GPT-4 and GPT-3.5 conversation representations in `only SFT` were intermingled. In contrast, `openchat-13b` clearly distinguished these representations according to their data sources, demonstrating the efficacy of our proposed C-RLFT in enriching input information.

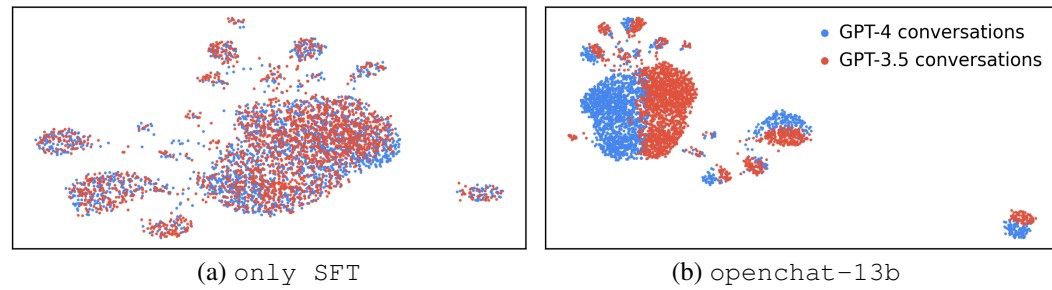

(a) `only SFT`                                        (b) `openchat-13b`

Figure 4: Visualization of GPT-4 and GPT-3.5 conversations' representations in `only SFT` and `openchat-13b`.

Secondly, given the significant impact of the class-conditioned policy, we further explore its effects on model performance during the inference phases by examining the influence of class-conditioned prompt tokens. In the inference phase, we use the GPT-4 prompt to induce `openchat-13b` to generate high-quality responses. Here we further verify the impacts of different inference prompts by replacing the GPT-4 prompt with the GPT-3.5 prompt. The comparison results, illustrated in Fig. 5, reveal a substantial performance decline when using the GPT-3.5 prompt instead of the GPT-4 prompt. This suggests that our `openchat-13b` model can distinguish the quality of different data sources based on our class-conditioned policy, and it further indicates that the representation space of GPT-4 is superior to that of GPT-3.5.

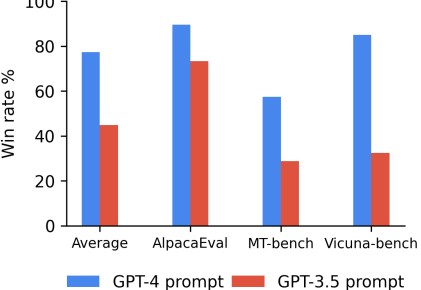

Figure 5: Effects of class-conditioned prompt tokens during inference phase.

## 5.4 EFFECTS OF DATA SIZE

In this section, we investigate the impact of varying data sizes on model performance. Specifically, we sub-sample one class in GPT-3.5 or GPT-4 with the ratio varying from 60% to 100% in 10% increments, while keeping the other class unchanged. It is worth noting that the total number of GPT-3.5 data is more than ten times larger than that of GPT-4.

The results are shown in Fig. 6. Firstly, we observe that the overall decline in both average performances is relatively modest, indicating that our `openchat-13b` is robust to variation in data size. Secondly, although the number of GPT-4 data points changes much less than GPT-3.5, the effect of varying GPT-4 data size is even more pronounced. This phenomenon demonstrates that expert data, while limited in quantity, is extremely important.

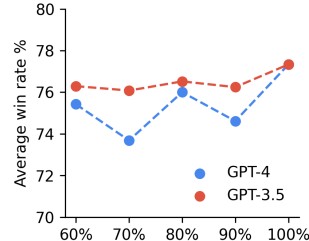

Figure 6: Effects of subsampling a specific class of data.

## 6 RELATED WORKS

**Large Language Models.** Recent years have witnessed significant advancements in LLMs, with models such as GPT-4 (OpenAI, 2023), PaLM (Chowdhery et al., 2022), and others comprising hundreds of billions or more parameters. This surge in LLMs extends beyond API-based models, as a suite of open-source language models like LLaMA (Touvron et al., 2023a), LLaMA-2 (Touvron et al., 2023b), and Falcon (Penedo et al., 2023) have emerged. This paper primarily focuses on the LLaMA base models, which are among the most popular open-source language models.

**Supervised Fine-tuning for LLMs.** A considerable body of work has been dedicated to enhancing large base language models through SFT. For instance, Alpaca (Taori et al., 2023) uses self-instruct (Wang et al., 2022) to generate 52k instruction-following demonstrations via `text-davinci-003`. This instruction data has been extensively applied in subsequent studies, such as Koala (Geng et al., 2023). Peng et al. (2023) follow Alpaca's setup but replace GPT-4 as the distillation teacher. WizardLM (Xu et al., 2023a) introduces Evol-Instruct, a technique that rewrites Alpaca's initial instruction data into more complex instructions, thereby enhancing the model's instruction-following capabilities. Other studies, such as UltraChat (Ding et al., 2023) and Baize (Xu et al., 2023b), have designed frameworks to obtain large-scale datasets of instructional conversations. Vicuna (Chiang et al., 2023), another popular variant, is the first to adopt ShareGPT with 70k user-shared ChatGPT conversations. Unlike previous SFT studies that treat all training data uniformly, we strive to maximize the use of mixed-quality data.

**Reinforcement Learning Fine-tuning for LLMs.** To better align with preferences beyond SFT, RLFT methods have been proposed. The most well-known method is RLHF (Ouyang et al., 2022), which involves collecting preference feedback from humans to train reward models. Subsequently, Proximal Policy Optimization (PPO) is used to train the target LLM to maximize the reward given. Most API-based LLMs, such as GPT-4, ChatGPT (OpenAI, 2023), and open-source models like Llama-2-chat series (Touvron et al., 2023b), utilize RLHF techniques. However, RLHF is a complex and unstable process that involves training a reward model and the LLM with an RL objective. As a result, simpler alternatives like DPO (Rafailov et al., 2023), RRHF (Yuan et al., 2023) have been proposed. DPO trains the LLM to predict and maximize reward simultaneously in a one-staged manner, while RRHF uses a ranking loss to encourage preferred answer output. Considering that the preference data is costly to collect, our method uses easily collectible and mixed-quality training data without any preference labels to finetune LLMs.

## 7 CONCLUSION AND FUTURE WORK

In this paper, we present OpenChat, an innovative framework featuring the Conditioned-RLFT method, tailored to advance open-source language models with mixed-quality data. Our model, `openchat-13b`, delivers the highest average performance on extensive benchmarks among all 13b open-source language models, demonstrating notable advantages such as simplicity, RL-free training, and minimal reward quality requirements. Despite these encouraging results, we acknowledge potential research areas for further improvement. Firstly, our assumption of different quality according to data sources may be overly simplistic, and the assigned coarse-grained rewards could be more finely tuned to reflect the actual quality of each data point. Secondly, while our model primarily focuses on enhancing instruction-following capabilities, exploring the application of OpenChat towards improving the reasoning abilities of LLMs offers a promising avenue for future work.

ETHICS STATEMENT

This study aims to advance open-source language models with mixed-quality data. Firstly, the proliferation of these open-source language models democratizes AI research by broadening its accessibility. These models serve as inclusive, transparent platforms that stimulate innovation and research. They cultivate a dynamic community of researchers from various backgrounds, thereby facilitating more comprehensive discussions and expedited collaborations. Secondly, refining the capability of these models to follow instructions can lead to more satisfying and safer responses, including the reduction of human bias and the promotion of fairness. Lastly, the dataset employed in this study consists of publicly available, user-shared instances, which are conveniently collected. This approach to data collection helps minimize potential data leakage and privacy concerns.

REPRODUCIBILITY STATEMENT

In line with our dedication to enhancing reproducibility, we have outlined our foundational model and training hyperparameters in Section 4.1. Besides, the training code, data and model weights for `openchat-13b` is publicly available at https://github.com/imoneoi/openchat and https://huggingface.co/openchat.

ACKNOWLEDGMENTS

The work is supported by the National Key R&D Program of China (2022ZD0160502) and the National Natural Science Foundation of China (No.61925601). Furthermore, we would like to thank Changling Liu in GPT Desk Pte. Ltd., Qiying Yu at Tsinghua University, Baochang Ma, and Hao Wan in 01.AI company for their resource support. We would like to express our gratitude to Jianxiong Li and Peng Li at Tsinghua University for their valuable discussion. Finally, we are also grateful to the developers of the following projects, which have contributed significantly to our research: Llama, self-instruct, FastChat (Vicuna), Alpaca, and StarCoder.

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

## A  EFFECTS OF CLASS-CONDITIONED PROMPT TOKENS

We further detect the effects of class-conditioned prompt tokens on model performance during the training phase. Our designed class-conditioned prompt tokens in different positions are shown in Table 3. we attempt three distinct initial prompt tokens in different positions: before speaker, before assistant, and beginning. The results are shown in Fig. 7. We observe that putting the conditioned prompt tokens in every turn (either before the speaker or before the assistant) performs similarly, but adding the condition only once at the beginning of the conversation performs much worse. This may be due to LLMs tend to forget the prompt at the beginning when the context is long or during subsequent turns. Touvron et al. (2023b) also observe the gradual loss of multi-turn consistency

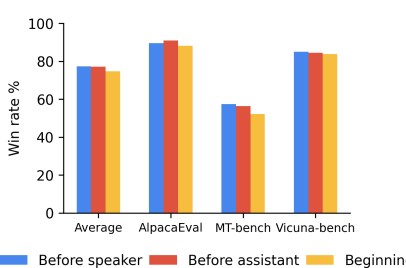

Figure 7: Effects of class-conditioned prompt tokens during training phase.

when the system prompt is put at the beginning. Therefore, we repeat the condition prompt every turn, to improve the effect of class-conditioned policy.

| Sources | Types | Conditioned Prompts |
|---|---|---|
| **GPT-4** | Before speaker | *GPT4 User:* Question`<|end_of_turn|>`*GPT4 Assistant:* |
| | Before assistant | *User:* Question`<|end_of_turn|>`*GPT4 Assistant:* |
| | Beginning | *Assistant is GPT4*`<|end_of_turn|>`*User:* Question`<|end_of_turn|>`*Assistant:* |
| **GPT-3.5** | Before speaker | *GPT3 User:* Question`<|end_of_turn|>`*GPT3 Assistant:* |
| | Before assistant | *User:* Question`<|end_of_turn|>`*GPT3 Assistant:* |
| | Beginning | *Assistant is GPT3*`<|end_of_turn|>`*User:* Question`<|end_of_turn|>`*Assistant:* |

Table 3: The attempted conditioned prompts during the training phase.

## B  DERIVATION OF THE OPTIMAL POLICY IN C-RLFT

The goal of C-RLFT is to find the optimal KL-regularized conditional policy. This optimization problem can be formulated as:

$$\max_{\pi} \mathbb{E}_{y\sim\pi}[r_c(x,y)] - \beta D_{\mathrm{KL}}(\pi, \pi_c) \tag{7}$$

To ensure $\pi$ is a valid probability distribution, we add the normalization constraint and the optimization problem becomes:

$$\max_{\pi} \mathbb{E}_{y\sim\pi}[r_c(x,y)] - \beta D_{\mathrm{KL}}(\pi, \pi_c) \tag{8}$$

$$s.t. \int_y \pi(y|x,c)dy = 1 \tag{9}$$

We can obtain the optimal solution of this constrained optimization problem by solving its Karush-Kuhn-Tucker (KKT) conditions. The Lagrangian of this problem is:

$$\mathcal{L}(\pi, \lambda) = \mathbb{E}_{y\sim\pi}[r_c(x,y)] - \beta D_{\mathrm{KL}}(\pi, \pi_c) + \lambda(1 - \int_y \pi(y|x,c)dy) \tag{10}$$

Following the KKT conditions, we take derivatives of $\mathcal{L}$ with respect to $\pi$ and $\lambda$, and set them to zero:

$$\frac{\partial \mathcal{L}}{\partial \pi} = r_c(x,y) + \beta \log \pi(y|x,c) - \beta \log \pi_c(y|x,c) + \beta - \lambda = 0 \tag{11}$$

$$\frac{\partial \mathcal{L}}{\partial \lambda} = 1 - \int_y \pi(y|x,c)dy = 0 \tag{12}$$

Solving these equations gives us the optimal policy $\pi^*$:

$$\pi^*(y|x,c) = \frac{1}{Z(x,c)} \pi_c(y|x,c) \exp\left(\frac{1}{\beta} r_c(x,y)\right) \tag{13}$$

$$Z(x,c) = \int_y \pi_c(y|x,c) \exp\left(\frac{1}{\beta} r_c(x,y)\right) dy \tag{14}$$

where $Z(x,c)$ is a normalization term ensuring that $\pi^*(y|x,c)$ is a valid probability distribution.

## C  DETAILS OF BENCHMARKS

This section provides the specifics of the benchmarks employed in our study:

- **AlpacaEval** (Li et al., 2023): This benchmark primarily assesses the model's ability to comprehend and execute user instructions. It incorporates a test set of 805 user instructions, collected from a diverse array of sources, and corresponding reference responses from `text-davinci-003`.
- **MT-bench** (Zheng et al., 2023): MT-bench presents a rigorous multi-turn benchmark designed to test both conversational and instruction-following capabilities. It includes 80 high-quality multi-turn questions that span eight distinct topics: writing, roleplay, extraction, reasoning, mathematics, coding, knowledge I (STEM), and knowledge II (humanities/social science).
- **Vicuna-bench** (Chiang et al., 2023): This benchmark evaluates the proficiency of large language models across eight question categories: generic, knowledge, roleplay, commonsense, Fermi problems, counterfactual scenarios, coding, mathematics, and writing.
- **AGIEval** (Zhong et al., 2023): AGIEval is a collection of human-centric standardized tests aimed at gauging the problem-solving abilities of language models. We include all English multiple-choice tasks in our evaluation, which encompass general college admission tests (SAT, AQuA-RAT), law school admission tests (LSAT), and civil service exams (LogiQA).

## D  MODEL INFORMATION

Table 4 presents the detailed specifications of the models used in our study, including the base models, context length, finetuning methods, and datasets employed.

| Model | Base Model | Context | Finetuning | Data |
|---|---|---|---|---|
| **Larger than 13b** | | | | |
| gpt-4 | - | 8k | SFT + RLFT | - |
| claude | - | 9k | SFT + RLFT | - |
| gpt-3.5-turbo | - | 4k | SFT + RLFT | - |
| llama-2-chat-70b | llama-2-70b | 4k | SFT + RLFT | ~27k high-quality SFT data + ~2.9M preference |
| guanaco-65b | llama-65b | 2k | SFT | ~9k OASST1 |
| guanaco-33b | llama-33b | 2k | SFT | ~9k OASST1 |
| **Equal to 13b** | | | | |
| vicuna-v1.1-13b | llama-13b | 2k | SFT | ~70k ShareGPT |
| wizardlm-v1.0-13b | llama-13b | 2k | SFT | ~70k gpt-3.5-turbo |
| ultralm-13b | llama-13b | 2k | SFT | ~1.5M UltraChat |
| llama-2-chat-13b | llama-2-13b | 4k | SFT + RLFT | ~27k high-quality SFT data + ~2.9M preference |
| vicuna-v1.5-13b | llama-2-13b | 4k | SFT | ~125k ShareGPT |
| wizardlm-v1.2-13b | llama-2-13b | 4k | SFT | ~250k gpt-3.5-turbo |
| **openchat-13b (ours)** | llama-2-13b | 4k | C-RLFT | ~70k ShareGPT |

Table 4: The details of the proposed OpenChat series models and other popular language models. The RLFT, SFT, and C-RLFT indicate reinforcement learning fine-tuning, supervised fine-tuning, and conditioned-RLFT proposed in our paper, respectively.

# E   EVALUATORS CONSISTENCY

While all benchmarks evaluate the agreement between humans and automatic evaluators, we also consider the self-enhancement bias as discussed by Zheng et al. (2023), where self-enhancement bias indicates that automatic evaluators may favor their own generated answers. To address this, we employ two additional automatic evaluators, `gpt-3.5` and `claude-2`, alongside the official evaluator `gpt-4`, to verify the consistency of evaluators on AlpacaEval. The results between `gpt-3.5` and `claude-2` are shown in Fig. 9, while the correlations between `gpt-4` and others ($r = 0.79$ and $0.73$) are detailed in App. H. The model performances evaluated by both evaluators show a strong Pearson correlation of $r = 0.91$. Most importantly, regardless of the automatic evaluator used, our `openchat-13b` outperforms all other 13b open-source language models, ranking within the top three among all API-based and open-source language models.

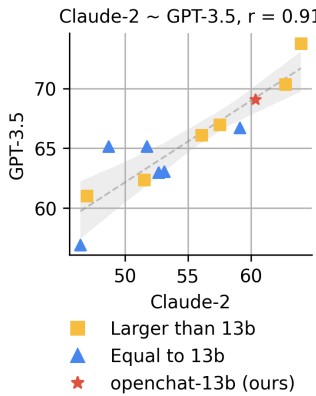

Figure 8: The consistency between GPT-3.5 and Claude-2 in AlpacaEval benchmark.

# F   AGIEVAL RESULTS

Table 5 presents the comprehensive results of AGIEval performance. All models are assessed using the official AGIEval zero-shot prompt and answer matching as described in Zhong et al. (2023). For conversation models (excluding `llama-2-13b`), we utilize the corresponding conversation templates and set the zero-shot prompt as the user's question.

| Task | openchat-13b | llama-2-13b | wizardlm-v1.2-13b | llama-2-chat-13b | vicuna-v1.5-13b |
|---|---|---|---|---|---|
| AQuA-RAT | 19.3 | 20.1 | **28.3** | 22.4 | 21.7 |
| LogiQA | **34.9** | 32.6 | 27.2 | 27.3 | 19.0 |
| LSAT-AR | 19.1 | 21.3 | **21.7** | 19.6 | 20.0 |
| LSAT-LR | **37.5** | 33.3 | 32.4 | 27.6 | 16.7 |
| LSAT-RC | 45.0 | **46.1** | 43.5 | 25.3 | 19.0 |
| SAT-English (w/o Psg.) | **44.7** | 41.3 | 33.5 | 25.2 | 26.7 |
| SAT-English | **66.0** | 65.5 | 65.5 | 25.7 | 28.2 |
| SAT-Math | **30.9** | 29.1 | 22.7 | 28.6 | 24.5 |
| Average | **36.4** | 35.0 | 32.8 | 25.8 | 20.7 |

Table 5: Zero-shot performance on AGIEval benchmark.

# G   CONVERSATION QUALITY SCORING

We employ `gpt-4` to automatically score the quality of conversations in the ShareGPT dataset on a scale of 1-10. For multi-turn conversations, each assistant's response is scored independently, taking into account the context of previous turns. The conversation's score is the average of all turn scores. The scoring prompt template is shown in Table 6.

| **Scoring prompt template** | [Instruction]
Please act as an impartial judge and evaluate the quality of the response provided by an AI assistant to the conversation displayed below. Your evaluation should consider factors such as the helpfulness, relevance, accuracy, depth, creativity, and level of detail of the response. Begin your evaluation by providing a short explanation. Be as objective as possible. After providing your explanation, you must rate the response on a scale of 1 to 10 by strictly following this format: "[[rating]]", for example: "Rating: [[5]]".

[Conversation]
{previous turns}

[The Start of Assistant's Response]
{response}
[The End of Assistant's Response] |
| --- | --- |

Table 6: Conversation evaluation prompts.

## H EVALUATORS CONSISTENCY

The correlations between arbitrary two models among `gpt-4`, `gpt-3.5` and `claude-2` are shown in Fig. 9.

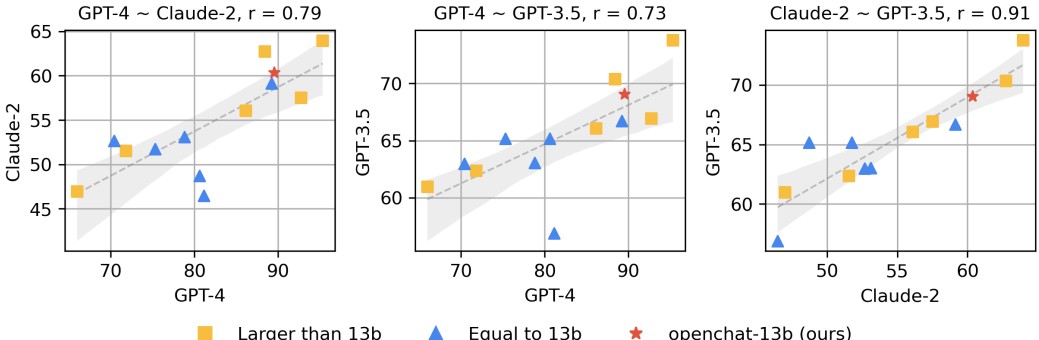

Figure 9: The consistency of three evaluators, GPT-4, GPT-3.5, Claude-2 in AlpacaEval benchmark.

# I CASE STUDY

| vicuna-13b-v1.5 | **[Turn #1] Craft an intriguing opening paragraph for a fictional short story. The story should involve a character who wakes up one morning to find that they can time travel.**
As the sun began to rise, Sarah stirred in her bed, the sound of birds chirping outside her window. But as she opened her eyes, she was not greeted by the familiar sight of her bedroom. Instead, she found herself standing in the middle of a bustling city street, the sound of cars and people rushing past her. Confused and disoriented, she looked around, trying to make sense of where she was and how she got there. And then she saw it - a shimmering aura around her, and a faint buzzing in her ears. It was then that she realized she had time traveled.
**[Turn #2] Summarize the story with three bullet points using only nouns and adjectives, without verbs.**
* Sarah wakes up in a bustling city street
* She discovers she has time traveled
* She is disoriented and confused |
|---|---|
| ultralm-13b | **[Turn #1] Craft an intriguing opening paragraph for a fictional short story. The story should involve a character who wakes up one morning to find that they can time travel.**
It was a typical morning when the alarm screeched through the room, jolting the sleeper awake. As they rubbed their eyes and reached for their glasses, they noticed something peculiar. The clock on the bedside table had stopped ticking. Yet, the room was bathed in a soft, golden light, and the air felt different. Suddenly, the sleeper felt a strange sensation in their chest, and before they knew it, they were hurtling through time.
**[Turn #2] Summarize the story with three bullet points using only nouns and adjectives, without verbs.**
Time Traveler:
- Unexpectedly travels through time
- Struggles to control powers
- Encounters historical figures and events |
| wizardlm-13b-v1.2 | **[Turn #1] Craft an intriguing opening paragraph for a fictional short story. The story should involve a character who wakes up one morning to find that they can time travel.**
It was just another ordinary day, a day like any other, or so it seemed. But for John, it was about to become the most extraordinary day of his life. As he groggily opened his eyes, he was hit with a wave of dizziness, a feeling he had never experienced before. Confused and disoriented, he rubbed his eyes, hoping it was just a side effect of his late night binge-watching of Doctor Who. But as he looked around, he realized that something was very wrong. His room seemed different, and the sunlight streaming in through the window was at an angle he had never seen before. It was then that he realized he had somehow managed to time travel.
**[Turn #2] Summarize the story with three bullet points using only nouns and adjectives, without verbs.**
* Character: John
* Time Travel
* Doctor Who
* Sunlight
* Room
* Day |
| llama-2-chat-13b | **[Turn #1] Craft an intriguing opening paragraph for a fictional short story. The story should involve a character who wakes up one morning to find that they can time travel.**
As I lay in bed, I felt a strange sensation wash over me, like the whisper of a breeze on a summer day. I opened my eyes to find myself in a world that was both familiar and yet, utterly foreign. The clock on my nightstand read 6:45 AM, but something felt off. The light pouring through the window seemed warmer, more golden than it should be. And then, I felt it - the sudden jolt of time travel. I blinked, and when I opened my eyes again, I was somewhere else entirely. The world outside my window was not the one I had left just moments before. I sat up, rubbing my eyes, trying to make sense of the strange sensation that had washed over me. How had I traveled through time? And where was I now? The possibilities were endless, and I couldn't wait to explore this new world and all its wonders.
**[Turn #2] Summarize the story with three bullet points using only nouns and adjectives, without verbs.**
Sure, here are three bullet points summarizing the story using only nouns and adjectives, without verbs:
* Character wakes up with strange sensation
* Golden light pours in through window
* Unfamiliar world outside with endless possibilities |
| **openchat-13b** | **[Turn #1] Craft an intriguing opening paragraph for a fictional short story. The story should involve a character who wakes up one morning to find that they can time travel.**
As the first rays of sunlight peeked through the window, Sarah closed her eyes, relishing the warmth that caressed her face. She never could resist the allure of a beautiful morning, but this particular day was about to become anything but ordinary. With a start, she sat up in bed, her heart pounding, as she realized she was somewhere else, somewhere she had never been before. Confused and exhilarated, Sarah discovered that she had the extraordinary ability to time travel, her very first morning blessed – or perhaps cursed – with the power to traverse the fabric of time.
**[Turn #2] Summarize the story with three bullet points using only nouns and adjectives, without verbs.**
*Sarah, a time traveler
*Mysterious morning adventure
*Discovery of extraordinary ability |

Table 7: Comparison of models on MT-bench. Only **openchat-13b** successfully adhered to the user's request in the second turn.

| Question | How long will it take to walk around the world, assuming that would be possible? |
|---|---|
| `vicuna-13b-v1.5` | It is not possible to walk around the world, as the distance around the globe is much greater than the distance across it. The circumference of the Earth at the equator is approximately 40,075 kilometers (24,901 miles), while the diameter is about 12,742 kilometers (7,918 miles).

Assuming you could walk continuously without rest, it would take approximately 1,900 years to walk around the world at an average walking speed of 5 kilometers (3.1 miles) per hour. However, this is just an estimate and does not take into account factors such as the terrain, altitude, and other conditions that would affect walking speed. |
| `ultralm-13b` | Walking around the world is not possible as the Earth is not a perfect sphere and has different land masses making it impossible to walk continuously without gaps. However, if we assume that there were no land masses and the Earth was a perfect sphere, it would take approximately 43,288.89 miles (69,617.05 kilometers) to walk around the equator. If we were to walk around the circumference of the Earth, it would take approximately 24,901.26 miles (40,074.91 kilometers). However, if we were to walk around the world in a straight line, it would take approximately 24,859.82 miles (40,008.06 kilometers) as the Earth is not a perfect sphere and has an elliptical shape. These distances are based on the equator circumference and the circumference of the Earth, respectively. The time it would take to walk around the world would depend on the individual's walking speed and other factors such as stops for rest and food. |
| `wizardlm-13b-v1.2` | As an AI, I'm assuming you're asking about walking around the Earth's equator. The distance around the Earth's equator at the equator itself is about 40,075 kilometers (24,901 miles). However, the actual distance would be much greater if you were to walk along a route that follows coastlines, navigates through mountains, and crosses other obstacles.

Now, let's consider that a person walks at a moderate pace of about 5 kilometers per hour. Walking around the world would then take approximately 8,015 hours or approximately 333,600 minutes.

To convert this into days, you would divide the total number of minutes by 24 (hours in a day): 333,600 minutes / 24 = 14,000 days.

So, if you were to walk around the world, it would take you approximately 14,000 days or approximately 38.89 years, assuming you walk continuously without rest and no obstacles. This is an estimate based on the distance around the Earth's equator and does not take into account real-world challenges like mountains, oceans, and other obstacles. |
| `llama-2-chat-13b` | Thank you for your question! While it is not possible to walk around the entire world, as there are many areas that are not accessible by foot, such as oceans and other bodies of water, it's important to note that even if it were possible, it would be a very long and challenging journey.

The circumference of the Earth at the equator is approximately 40,075 kilometers (24,901 miles). However, it's important to consider that walking is not a viable mode of transportation for long distances, as it would be extremely time-consuming and would require a tremendous amount of physical endurance.

Furthermore, it's important to note that walking through certain areas of the world, such as deserts or other remote regions, could be dangerous due to harsh weather conditions, lack of resources, and other hazards.

Therefore, it's not possible to provide an accurate estimate of how long it would take to walk around the world, as it is not a feasible or safe journey. It's important to prioritize safe and accessible forms of transportation when traveling, and to plan your journeys with careful consideration of your physical limitations and the resources available to you. Is there anything else I can help with? |
| **`openchat-13b`** | It is possible to walk around the world, though it would be a continuous, grueling journey that would take several years, if not a decade or more, depending on the route chosen, individual physical capabilities, and other factors. The distance around the world varies depending on the route, but a commonly used measure is the equatorial circumference of the Earth, which is approximately 24,901.48 miles (40,075.01 kilometers).

Walking at a moderate pace of around 3 miles (4.8 kilometers) per hour, it would take an average person nearly 3 years and 3 months to walk around the world (assuming no breaks or rest days). However, this is just an estimate, and the actual time required would depend on various factors, such as:

1. Route: The distance around the world varies depending on the route taken. For example, the shortest distance is along the Equator, while the longest distance is along the meridians of longitude.
2. Terrain: Walking through mountainous or heavily forested areas may be slower and more challenging than walking on flat terrain.
3. Weather: Adverse weather conditions, such as extreme heat or cold, high winds, or heavy rainfall, can make walking more difficult and slow down progress.
4. Fitness and endurance: An individual's physical fitness, endurance, and ability to withstand the rigors of constant walking will affect their pace and overall journey time.
5. Nutrition and rest: Proper nutrition and adequate rest are essential for maintaining physical and mental stamina. Taking regular breaks and rest days will prolong the journey.

Keep in mind that walking around the world is an extraordinary feat that would require extensive preparation, both physically and logistically. It's essential to consider the many challenges and potential hazards involved before embarking on such an adventure. |

Table 8: Comparison of models on AlpacaEval benchmark. Specifically, the models `ultralm-13b` and `llama-2-chat-13b` fail to provide a direct answer, while `vicuna-13b-v1.5` and `wizardlm-13b-v1.2` offer estimates with significant discrepancies. **`openchat-13b`** provides a reasonable estimate, demonstrating the varying capabilities of models in addressing complex hypothetical questions.

| Question | Given that $f(x) = 5x^3 - 2x + 3$, find the value of f(2). |
|---|---|
| `vicuna-13b-v1.5` | To find the value of f(2), we simply substitute 2 for x in the expression for f(x): 
 $f(2) = 5(2)^3 - 2(2) + 3$ 
 Simplifying this expression, we get: 
 $f(2) = 25 - 4 + 3$ 
 $f(2) = 24$ 
 Therefore, the value of f(2) is 24. |
| `ultralm-13b` | To find the value of f(2), we substitute 2 for x in the function f(x): 
 $f(2) = 5(2)^3 - 2(2) + 3$ 
 $f(2) = 20 - 4 + 3$ 
 $f(2) = 25$ 
 Therefore, the value of f(2) is 25. |
| `wizardlm-13b-v1.2` | The problem is asking us to find the value of the function f(2). The function is given as $f(x) = 5x^3 - 2x + 3$. 
 Step 1: Substitute the value of x into the function 
 We are told to substitute the value of x with 2. So, we replace every x in the function with 2. This gives us: 
 $f(2) = 5(2)^3 - 2(2) + 3$ 
 Step 2: Simplify the expression 
 Next, we simplify the expression. The expression $5(2)^3$ simplifies to 5*8, which equals 40. The expression 2(2) simplifies to 4. And the number 3 remains the same. So, we have: 
 $f(2) = 40 - 4 + 3$ 
 Step 3: Perform the subtraction and addition 
 Now, we perform the subtraction and addition in the correct order. First, we subtract 4 from 40, which gives us 36. Then, we add 3 to 36, which gives us 39. So, we have: 
 $f(2) = 39$ 
 So, the value of f(2) is 39. |
| `llama-2-chat-13b` | Sure! To find the value of f(2), we can plug 2 into the function $f(x) = 5x^3 - 2x + 3$. 
 $f(2) = 5(2)^3 - 2(2) + 3$ 
 $= 5(8) - 2(2) + 3$ 
 $= 5(8) - 4 + 3$ 
 $= 40 - 4 + 3$ 
 $= 36$ 
 So, f(2) = 36. |
| **`openchat-13b`** | To find the value of f(2), we need to evaluate the function when x = 2. 
 $f(2) = 5(2)^3 - 2(2) + 3$ 
 $f(2) = 5(8) - 4 + 3$ 
 $f(2) = 40 - 4 + 3$ 
 $f(2) = 39$ |

Table 9: Comparison of models on Vicuna-bench. Only **openchat-13b** and `wizardlm-13b-v1.2` provide the correct answer to this math problem.

