# OpenReview forum: "OpenChat: Advancing Open-source Language Models with Mixed-Quality Data"
_ICLR.cc/2024/Conference — ICLR 2024 poster_

### Official Review · Reviewer_g6Xd · 2023-10-31

**Soundness:** 3 good
**Presentation:** 3 good
**Contribution:** 3 good
**Rating:** 6
**Confidence:** 3

**Summary:**

The paper presents OpenChat, a framework that uses a conditioned-RLFT (C-RLFT) method to improve open-source language models with mixed-quality instruction tuning data. The authors demonstrate that their model, openchat-13b, outperforms other 13b open-source language models on extensive benchmarks, with notable advantages such as simplicity, RL-free training, and minimal reward quality requirements.

**Strengths:**

* The proposed method, conditioned-RLFT (C-RLFT), is overall novel and simple. It regards different data sources as coarse-grained reward labels and learns a class-conditioned policy to leverage complementary data quality information. The optimal policy in C-RLFT can be solved through single-stage, RL-free supervised learning, which is lightweight and avoids costly human preference labeling.
* The authors demonstrate that OpenChat achieves the highest average performance among all 13b open-source language models across commonly used instruction-following benchmarks.

**Weaknesses:**

* My biggest concern is that while the paper claims their approach to be based on single-stage RL-free supervised learning, they do use data generated by API models (GPT-4 & GPT-3.5), and these models are believed to have gone through multi-stage training with RLHF. If the C-RLFT method leverages these kinds of data for their training procedure, it's not very convincing to me that the method really works in a fundamentally different way from prior approaches (SFT + RLHF). It also makes the comparisons to baselines not fully fair, since many baselines (under the 13b model scale) being compared to have not leveraged training data generated by (much larger) API models.

**Questions:**

* Would the proposed method work if the mixed-quality data are generated by open-sourced (and potentially smaller) models, such as LLaMA-13b vs LLaMA-7b?
* I'd like to see a discussion of Xu et al. (I understand that Xu et al. came out probably after this work, but it seems to me that there are some similarities in methodology between the two papers).

Reference:
Xu et al. “Contrastive Post-training Large Language Models on Data Curriculum.” ArXiv abs/2310.02263

---

> ### Author Response · Authors · 2023-11-19
> **Response to Weakness 1**
>
> > **[W1]** My biggest concern is that while the paper claims their approach to be based on single-stage RL-free supervised learning, they do use data generated by API models (GPT-4 & GPT-3.5), and these models are believed to have gone through multi-stage training with RLHF. If the C-RLFT method leverages these kinds of data for their training procedure, it's not very convincing to me that the method really works in a fundamentally different way from prior approaches (SFT + RLHF). It also makes the comparisons to baselines not fully fair, since many baselines (under the 13b model scale) being compared to have not leveraged training data generated by (much larger) API models.
>
> **Response:**
>
> - Thanks for your question. Here we summarized the fine-tuning data sources of OpenChat and other popular open-source language models. Actually, most open-source SFT models leverage training data generated by (much larger) API models, except for `llama-2-chat-13b`. It should be noted many existing models have more stringent requirements on the finetuning data, either data quantity or data quality. By contrast OpenChat enables to use the minimal amount of potentially expensive expert data as well as more easily obtainable medium-quality data to achieve improved LLM finetuning performance.
>
>
> | Model                     | Base Model    | Finetuning | Data                                             |
> | ------------------------- | ------------- | ---------- | ------------------------------------------------ |
> | **Equal to 13b**          |               |            |                                                  |
> | `vicuna-v1.1-13b`         | `llama-13b`   | SFT        | ~70k ShareGPT (GPT-4 + GPT-3.5)                  |
> | `wizardlm-v1.0-13b`       | `llama-13b`   | SFT        | ~70k GPT-3.5                                     |
> | `ultralm-13b`             | `llama-13b`   | SFT        | ~1.5M UltraChat (GPT-3.5)                        |
> | `llama-2-chat-13b`        | `llama-2-13b` | SFT + RLFT | ~27k high-quality SFT data + ~2.9M preference    |
> | `vicuna-v1.5-13b`         | `llama-2-13b` | SFT        | ~125k ShareGPT (GPT-4 + GPT-3.5)                 |
> | `wizardlm-v1.2-13b`       | `llama-2-13b` | SFT        | ~250k Evol-Instruct + ShareGPT (GPT-4 + GPT-3.5) |
> | **`openchat-13b (ours)`** | `llama-2-13b` | C-RLFT     | ~70k ShareGPT (GPT-4 + GPT-3.5)

---

> ### Author Response · Authors · 2023-11-19
> **Response to Question 1 and 2**
>
> > **[Q1]** Would the proposed method work if the mixed-quality data are generated by open-sourced (and potentially smaller) models, such as LLaMA-13b vs LLaMA-7b?
>
> **Response:**
>
> - Thanks for your comment. It's not appropriate to train the base model (`llama-2-13b`) on datasets generated with models weaker than the base model (`LLaMA-7b`, `LLaMA-13b`), as they have worse capabilities and are unlikely to provide useful information for model fine-tuning.
> - However, if the reviewer wants to look into the performance of C-RLFT on datasets other than ShareGPT (GPT-4 + GPT-3.5), we have added an experiment using the OpenOrca dataset [1]. OpenOrca is a reasoning dataset that is generated by human-written and GPT-4. We treat human data as $\mathcal{D}_{exp}$ and GPT-4 data as $\mathcal{D}_{sub}$, respectively. The results are shown in the following Table, where C-RLFT outperforms SFT on both datasets as well as the combined dataset.
>
> | Type                   | BBH      |
> | ---------------------- | -------- |
> | C-RLFT (Human + GPT-4) | **49.4** |
> | SFT (Human + GPT-4)    | 47.9     |
> | SFT (Human)            | 47.1     |
> | SFT (GPT-4)            | 46.3     |
>
> ---
>
> > **[Q2]** I'd like to see a discussion of Xu et al. (I understand that Xu et al. came out probably after this work, but it seems to me that there are some similarities in methodology between the two papers).
>
> **Response:**
>
> Thanks for your question. Our C-RLFT is fundamentally different from Xu et al., as their proposed method is still a preference-based method, like RLHF, while our method simply uses two different quality datasets.
>
> - Our C-RLFT method has extremely relaxed requirements for the training dataset. It only requires to have a small set of expert data and a larger set of medium-quality data, without the need for preference labels. Meanwhile, C-RLFT reformulates the RL objective to a weighted supervised loss, which is more stable compared to the RL training in existing RLFT methods.
> - Xu et al., like all RLHF methods, require pairwise preference labels for every question $x_i \mapsto (y_i^+, y_i^-)$, i.e., for every question $x_i$, it needs a preferred answer $y_i^+$ that is better than the worse answer $y_i^-$. Moreover, Xu et al. need to use two different quality LLMs to generate positive and negative answers, and then run DPO for final policy fine-tuning. The entire learning process is very heavy and inherits the potential instability and hyperparameter sensitivity of DPO-style RL training. Moreover, the positive and negative answers generated by LLMs could also introduce potential model-specific biases to the preference data. Nevertheless, we have cited this paper in our revision. We thank the reviewer for providing this reference.
>
> **References:**
>
> [1] Orca: Progressive Learning from Complex Explanation Traces of GPT-4. arXiv 2023.

---

> > ### Comment · Reviewer_g6Xd · 2023-11-23
> > **Response to Authors**
> >
> > I thank the authors for their response. I'm keeping my score since it's already positive.

---

### Official Review · Reviewer_yFnS · 2023-11-05

**Soundness:** 2 fair
**Presentation:** 3 good
**Contribution:** 3 good
**Rating:** 6
**Confidence:** 3

**Summary:**

This study proposes OpenChat, a new framework that refines open-source language models using mixed-quality data, applying a unique C(onditioned)-RLFT method that bypasses the necessity for detailed preference labels. The proposed method allows for a more straightforward policy learning process, leading to a large improvement in model performance. The openchat-13b model, fine-tuned via C-RLFT, notably surpasses peers in average performance across benchmarks and exhibits superior generalization in AGIEval tests. The researchers plan to make their code, data, and models publicly accessible, facilitating advancements in the field.

**Strengths:**

The proposed method is emphasizing data quality, which is a critical aspect of the model alignment problem.
The framework is straightforward and replicable, delivering significant improvements across various benchmark datasets.
The evaluation is comprehensive, conducted against several established models and on a wide range of benchmark datasets, with the experimental design being thoroughly considered.

**Weaknesses:**

The paper does not adequately address the challenge of estimating data quality when leveraging mixed-quality data sources. The method presented seems to oversimplify this estimation and the associated reinforcement learning (RL) reward mechanism:

The process of data quality estimation may be too simplified and not easily generalizable to realistic settings involving human feedback. The estimation relies on the assumption that GPT-4 provides better quality outputs than GPT-3.5, and GPT-3.5 surpasses the base model. Although this may be true in some instances, it's not universally applicable across all domains and might not hold as the base model evolves. This assumption could be problematic if the alignment is based on a superior base model or a larger one. Furthermore, presuming that the GPT series invariably delivers superior samples can inadvertently lead to the replication of any negative behaviors from these models, making it challenging to improve upon them with base model developments.

The method applies static, positive, example-level rewards for learning, foregoing the primary advantages of an RL framework. An RL approach can benefit from both positive and negative samples and can assign more nuanced, token-level rewards, rather than broad example-level ones. It also enables the formulation of objectives that are not readily translated into intermediate scores. The static, example-level rewards could potentially be reformulated into a supervised learning loss, combined with data weighting strategies. The supervised learning loss with data weighting strategies are

**Questions:**

1.  One of interesting question when handling the mixed data quality problem is on the contribution of data with various quality. In this paper, one research question is what the performance will be if only GPT-4 samples are used even the size is ten times smaller than GPT-3.5.  As a follow-up, how does the quality of data from different versions of language models, like GPT-4 and GPT-3.5, impact the performance of fine-tuned models, and what is the optimal quantity ratio of high-quality to lower-quality data necessary for achieving equivalent performance enhancements?

2. What strategies can be implemented within the proposed framework to mitigate the transfer of potentially negative behaviors from GPT-4 examples to the base model, and how can such behaviors be identified and counteracted effectively?

3. In what ways can the proposed framework be adapted to more complex and realistic data environments, such as those involving human-generated data, which may exhibit significant variations in quality and where quality assessments may be more subjective?

4. How does a supervised learning approach with data weighting, predicated on the assumption that GPT-4 provides higher quality data than GPT-3.5, compare in performance to the RL framework, and can the simplicity of supervised loss be reconciled with the advantages of an RL-based approach?

---

> ### Author Response · Authors · 2023-11-19
> **Response to Weakness 1 and Question 3**
>
> > **[W1]** The paper does not adequately address the challenge of estimating data quality when leveraging mixed-quality data sources. The method presented seems to oversimplify this estimation and the associated reinforcement learning (RL) reward mechanism:
> The process of data quality estimation may be too simplified and not easily generalizable to realistic settings involving human feedback. The estimation relies on the assumption that GPT-4 provides better quality outputs than GPT-3.5, and GPT-3.5 surpasses the base model. Although this may be true in some instances, it's not universally applicable across all domains and might not hold as the base model evolves. This assumption could be problematic if the alignment is based on a superior base model or a larger one. Furthermore, presuming that the GPT series invariably delivers superior samples can inadvertently lead to the replication of any negative behaviors from these models, making it challenging to improve upon them with base model developments.
> >
> > **[Q3]** In what ways can the proposed framework be adapted to more complex and realistic data environments, such as those involving human-generated data, which may exhibit significant variations in quality and where quality assessments may be more subjective?
>
> **Response:**
> - First, our proposed C-RLFT is not designed to "estimate data quality" as in typical RLFT methods, but rather provide the cheapest solution to bypass this step in LLM fine-tuning. In typical RLFT methods, we have to use lots of expensive human preference data to learn reward models, in order to explicitly estimate data quality. This poses major obstacles and instability in LLM fine-tuning, as human preference collection is very costly and the two-stage training process (learn reward first and then run RL algorithm like PPO) can be very unstable. By contrast, our method smartly avoids the explicit data quality estimation problem by simply leveraging the quality difference between two datasets to achieve the same policy fine-tuning goal.
> - Second, our proposed C-RLFT is not bounded with GPT-4 and GPT-3.5 data. As clearly described in our paper, our method only needs an expert dataset $\mathcal{D}\_{exp}$ and a sub-optimal dataset $\mathcal{D}\_{sub}$. The GPT-4 and GPT-3.5 datasets are only for demonstrative purposes, and in principle, one could use any expert & sub-optimal datasets to fine-tune their models. To further address the concern of the reviewer, we conducted additional experiments to tune our model with human-written data (treated as expert data) + GPT-4 data (treated as sub-optimal data) from the OpenOrca dataset [1], the results are presented as follows. Our proposed C-RLFT still outperforms SFT models trained with Human or GPT-4 data, or a combination of both.
>
> Table: Comparison of C-RLFT and SFT on Human & GPT-4 datasets (we report the accuracy of Big-Bench-Hard (BBH) as used in Orca[1] for comparison).
> | Type                   | BBH      |
> | ---------------------- | -------- |
> | C-RLFT (Human + GPT-4) | **49.4** |
> | SFT (Human + GPT-4)    | 47.9     |
> | SFT (Human)            | 47.1     |
> | SFT (GPT-4)            | 46.3     |

---

> ### Author Response · Authors · 2023-11-19
> **Response to Question 1**
>
> > **[Q1]** One of interesting question when handling the mixed data quality problem is on the contribution of data with various quality. In this paper, one research question is what the performance will be if only GPT-4 samples are used even the size is ten times smaller than GPT-3.5. As a follow-up, how does the quality of data from different versions of language models, like GPT-4 and GPT-3.5, impact the performance of fine-tuned models, and what is the optimal quantity ratio of high-quality to lower-quality data necessary for achieving equivalent performance enhancements?
>
> **Response:**
>
> - Thanks for the reviewer’s comment. It should be noted that if only GPT-4 samples are used, then C-RLFT actually reduces to perform SFT on a single source of data. To fully address the reviewer’s concern, we conducted additional experiments to compare C-RLFT on GPT-4 + GPT-3.5 against SFT methods tuned on GPT-4 and GPT-3.5, as well as their combination. The results are presented in the following table.
>
> | Type                     | Average  | AlpacaEval | MT-bench | Vicuna-bench |
> |--------------------------|----------|------------|----------|--------------|
> | C-RLFT (GPT-4 + GPT-3.5) | **77.3** | **89.5**   | **57.5** | **85.0**     |
> | SFT (GPT-4)              | 67.9     | 85.8       | 33.4     | 84.4         |
> | SFT (GPT-4 + GPT-3.5)    | 52.7     | 78.6       | 33.1     | 46.3         |
> | SFT (GPT-3.5)            | 42.8     | 76.5       | 16.9     | 35.0         |
>
>
> - Regarding the question of the reviewer on the impact of training C-RLFT from different versions of the language model, please refer to our additional results on C-RLFT with Human + GPT-4 data in our response to **W1**. It can be shown that as long as we can have two datasets with different quality levels, C-RLFT can still achieve better performance as compared to performing SFT on such datasets.
> - Regarding the optimal quantity ratio of high-quality to low-quality data, we actually have ablation results (Figure 7) in our main paper. We sub-sample one class in GPT-3.5 or GPT-4 with the ratio varying from 60% to 100%, while keeping the other class unchanged. The results show that the average performance is relatively stable, demonstrating that our method is robust to variation in dataset ratios, as long as the expert data size is not overly small. For details, please check Section 5.5 in our paper.

---

> > ### Comment · Reviewer_yFnS · 2023-11-19
> > **Follow-up Discussion**
> >
> > Thanks for adding the experiments. However, there's some confusion regarding why C-RLFT reduces to SFT when solely employing GPT-4 samples. RLFT remains applicable with GPT-4 data by considering all samples with r_c(x,y) as 1. As the authors highlighted, C-RLFT differs from standard supervised learning, focusing on a KL-regularized RL problem instead. Therefore, for a proper evaluation of data quality, it seems more appropriate to conduct an ablation study on the data alone, rather than altering the objective function.

---

> ### Author Response · Authors · 2023-11-19
> **Response to Question 2**
>
> > **[Q2]** What strategies can be implemented within the proposed framework to mitigate the transfer of potentially negative behaviors from GPT-4 examples to the base model, and how can such behaviors be identified and counteracted effectively?
>
> **Response:**
> It should be noted that our proposed C-RLFT framework is not bounded with GPT-4 and GPT-3.5 data. If the user is concerned with potential negative behaviors in the GPT-4 examples, one can simply replace the expert dataset $\mathcal{D}\_{exp}$ with a small set of carefully curated human expert data, like used in LIMA [4].

---

> ### Author Response · Authors · 2023-11-19
> **Response to Question 4**
>
> > **[Q4]** How does a supervised learning approach with data weighting, predicated on the assumption that GPT-4 provides higher quality data than GPT-3.5, compare in performance to the RL framework, and can the simplicity of supervised loss be reconciled with the advantages of an RL-based approach?
>
> **Response:**
>
> - Note that C-RLFT is not a typical supervised learning approach, but solving a KL-regularized RL problem: $J\_\text{C-RLFT}(\theta)=\mathbb{E}\_{y \sim \pi\_\theta}[r\_c(x,y)]-\beta D\_{\mathrm{KL}}\left(\pi\_\theta, \pi\_c\right)$. Its optimal policy can extracted using a weighted supervised learning loss, but its nature is RL.
> - Additionally, if the reviewer checks the `llama-2-chat-13b` baseline in the paper, it has the same base model as our `openchat-13b`, but is tuned using SFT+RLFT with ~27k high-quality SFT data + ~2.9M preference, but its average score is still worse than our `openchat-13b`. This shows the strength of our proposed C-RLFT.
>
> | Models             | Base Models   | Method     | Dataset                                       | AlpacaEval | MT-bench | Vicuna-bench | Average  |
> | ------------------ | ------------- | ---------- | --------------------------------------------- | ---------- | -------- | ------------ | -------- |
> | `llama-2-chat-13b` | `llama-2-13b` | SFT + RLFT | ~27k high-quality SFT data + ~2.9M preference | 81.1       | 55.3     | **86.9**     | 74.4     |
> | `openchat-13b`     | `llama-2-13b` | C-RLFT     | ~70k ShareGPT                                 | **89.5**   | **57.5** | 85.0         | **77.3** |

---

> ### Author Response · Authors · 2023-11-19
> **References**
>
> **References:**
>
> [1] Orca: Progressive Learning from Complex Explanation Traces of GPT-4. arXiv 2023.
>
> [2] Direct Preference Optimization: Your Language Model is Secretly a Reward Model. NeurIPS 2023.
>
> [3] Learning from Human Feedback without RL. arXiv 2023.
>
> [4] LIMA: less is more for alignment. arXiv 2023.

---

> ### Author Response · Authors · 2023-11-20
> **Response to follow-up discussion**
>
> > Thanks for adding the experiments. However, there's some confusion regarding why C-RLFT reduces to SFT when solely employing GPT-4 samples. RLFT remains applicable with GPT-4 data by considering all samples with r_c(x,y) as 1. As the authors highlighted, C-RLFT differs from standard supervised learning, focusing on a KL-regularized RL problem instead. Therefore, for a proper evaluation of data quality, it seems more appropriate to conduct an ablation study on the data alone, rather than altering the objective function.
>
> **Response:**
>
> Thank you for your comments. It seems there is some confusion regarding the reduction form of C-RLFT on the single dataset setting. We provide detailed clarification as follows:
> - First, if we only have a single quality dataset, e.g., GPT-4, then the class-condition policy $\pi_{\theta}(\cdot|x,c)$ of C-RLFT will reduce to a non-conditioning policy $\pi_{\theta}(\cdot|x)$ since the conditioning term $c$ will remain constant.
> - Second, the class-conditioned rewards $r_c(x,y)$ also becomes constant (denoted as $r$) since there is no quality difference in the data.
> - Under such cases, the final form of the extracted optimal policy in Eq.(6) in our paper will become:
> $$\pi\_{\theta}=\arg\max\_{\theta}\mathbb{E}\_{x,y,c\sim \mathcal{D}\_c}\left[\exp\left(r\_c(x,y)/\beta\right)\log\pi\_{\theta}(y|x,c)\right]$$
> $$=\arg\max\_{\theta}\mathbb{E}\_{x,y\sim \mathcal{D}\_c}\left[\exp\left(r/\beta\right)\log\pi\_{\theta}(y|x)\right]=\arg\max\_{\theta}C\cdot\mathbb{E}\_{x,y\sim \mathcal{D}\_c}\left[\log\pi\_{\theta}(y|x)\right]$$
> where $C$ is a constant and the optimal policy extraction becomes exactly SFT on the single dataset $\mathcal{D}_c$ (e.g., GPT-4 dataset).
> - To further address the reviewer's concern, we ran C-RLFT on 8x A100 80G GPUs with only the GPT-4 dataset, using the same base model (`llama-2-13b`) and hyperparameters as in our paper. The table below presents the results, demonstrating that C-RLFT and SFT exhibit similar performance on GPT-4 data, which supports our explanation. Moreover, C-RLFT with both GPT-3.5 and GPT-4 data outperforms the models trained with GPT-4 data alone by a significant margin, indicating that the model effectively learns from the quality difference between datasets.
>
> | Model                    | Average  | AlpacaEval | MT-bench | Vicuna-bench |
> | ------------------------ | -------- | ---------- | -------- | ------------ |
> | C-RLFT (GPT-4 + GPT-3.5) | **77.3** | **89.5**   | **57.5** | **85.0**     |
> | C-RLFT (GPT-4 only)      | 67.5     | 86.9       | 33.1     | 82.5         |
> | SFT (GPT-4)              | 67.9     | 85.8       | 33.4     | 84.4         |

---

> > ### Comment · Reviewer_yFnS · 2023-11-23
> > **Thanks for follow-up experiments**
> >
> > Thank you to the authors for conducting additional experiments.
> >
> > A point of consideration is the current methodology of manually setting reward scores for GPT-4 and GPT-3.5 data. I suggest conducting further studies to understand the impact of different reward values. This exploration could provide more insights into the optimal setting of these parameters.
> >
> > Another area of concern is the disparity between data generated by GPT models and data curated by humans. The patterns in model-generated data tend to be more uniform and easier to learn, in contrast to the diverse and complex nature of human-curated data. This difference poses a challenge in generalizing conclusions drawn from a knowledge distillation setting to scenarios involving human data. The distinction between these two types of data should not be overlooked or underestimated.
> >
> > While these have alleviated my concerns regarding the data aspect, I still have reservations about the use of a fixed reward in the reinforcement learning framework. Consequently, I have adjusted my score upwards.

---

> > > ### Author Response · Authors · 2023-11-23
> > > **Thanks for your response**
> > >
> > > Thank you for your valuable feedback and for adjusting the score upwards. Indeed, the insightful points you raised align perfectly with the areas we are currently working to improve.
> > >
> > > - **Manual setting of reward scores**: We are in agreement with your suggestion. In our future work section, we have discussed fine-tuning the assigned coarse-grained rewards to better reflect the actual quality of each data point. We plan to conduct an analysis study to understand the impact of different reward values in our next version. Additionally, we are exploring new methods to improve this aspect and will release it soon.
> > >
> > > - **Disparity between data generated by GPT models and data curated by humans**: As per your suggestion, we have further compared C-RLFT and SFT on human and GPT-4 datasets in our first response. The experimental results indicate that our C-RLFT consistently performs better. Furthermore, as you rightly pointed out, we found a difference in the distribution of manually constructed data and GPT-4 data, with the manually curated data being more straightforward and simple. We will further explore it more clearly.
> > >
> > > We appreciate your constructive feedback and look forward to incorporating your suggestions into our ongoing work.

---

### Official Review · Reviewer_U8fK · 2023-11-07

**Soundness:** 3 good
**Presentation:** 4 excellent
**Contribution:** 2 fair
**Rating:** 6
**Confidence:** 3

**Summary:**

This paper proposes OpenChat, a framework for fine-tuning open-source language models with mixed-quality data. It proposes Conditioned-RLFT: A novel method that leverages coarse-grained rewards and class-conditioned policies to align the language model with human goals. This method is simple, RL-free, and does not require costly human feedback. Using RLFT, they finetuned OpenChat-13b, a language model based on LLaMA-2-13b and the ShareGPT dataset, which consists of conversations from GPT-4 and GPT-3.5. This model achieves the highest average performance on three standard benchmarks for instruction following ability among all 13b open-source language models.

**Strengths:**

- Originality: The paper introduces a new method, C-RLFT, that leverages coarse-grained rewards and class-conditioned policies to align the language model with human goals. The paper also provides a theoretical analysis and derivation of the optimal policy for C-RLFT. The resulting framework is a conditioned SFT with weighted loss, which is easy to implement and more stable than RLHF.
- Quality: The paper presents evaluation results on three benchmarks to assess instruction following ability of the proposed model. The baselines include popular open and closed LLMs. The paper also performs ablation studies, representation visualization, prompt token effects, evaluator consistency, and data size effects to validate the effectiveness and robustness of OpenChat.
- Clarity: The paper is well-written and organized.
- Significance: The paper addresses an important and challenging problem of fine-tuning open-source language models with mixed-quality data. The paper demonstrates that OpenChat can achieve superior performance and generalization with simple and RL-free training, minimal reward quality requirements, and easily collectible data.

**Weaknesses:**

- The paper claims the superiority of C-RLFT over RLFT. But the claim is only supported by openchat-13b, which distilled its knowledge from two RLFT models: GPT-4 and GPT-3.5. So, it is only fair to say C-RLFT is better than SFT when distilling from GPT models, which is based on the fact the openchat-13b is better than vicuna-13b-1.5. However, one cannot say C-RLFT is better than RLFT, because openchat-13b learns from models trained by RLFT.
- It is not clear if the "low quality data" is even useful at all. (see my question below)

**Questions:**

- What is the performance if the llama model is only trained on GPT-4 data? I feel it should be another ablated model to demonstrate the value of "low quality data".

---

> ### Author Response · Authors · 2023-11-19
> **Response to Weakness 1**
>
> > **[W1]** The paper claims the superiority of C-RLFT over RLFT. But the claim is only supported by openchat-13b, which distilled its knowledge from two RLFT models: GPT-4 and GPT-3.5. So, it is only fair to say C-RLFT is better than SFT when distilling from GPT models, which is based on the fact the openchat-13b is better than vicuna-13b-1.5. However, one cannot say C-RLFT is better than RLFT, because openchat-13b learns from models trained by RLFT.
>
> **Response:**
> We thank the reviewer for this thoughtful comment. Regarding the reviewer's concern, we provide detailed responses as follows:
>
> - First, we'd like to clarify that our model is not simply distilling from GPT models. We have added the following experiments to compare C-RLFT vs SFT on different GPT data.
> As the results show, if we simply distill a pretrained model by performing SFT on GPT-4 or GPT-3.5 data, their performances are much worse as compared to our proposed C-RLFT. Moreover, SFT with both GPT-4 and GPT-3.5 (\~70k) performs even worse than SFT tuned with only GPT-4 (\~6k), suggesting that indiscriminatively training with mixed quality data is likely to negatively impact learning (also discussed in our paper). By contrast, C-RLFT can smartly leverage the quality contrast of two datasets and achieve much better performance, this cannot be explained by distillation from better GPT models.
>
> Table R1: Comparative performance of C-RLFT and SFT tuned on different data sources.
>
> | Type                     | Average  | AlpacaEval | MT-bench | Vicuna-bench |
> |--------------------------|----------|------------|----------|--------------|
> | C-RLFT (GPT-4 + GPT-3.5) | **77.3** | **89.5**   | **57.5** | **85.0**     |
> | SFT (GPT-4)              | 67.9     | 85.8       | 33.4     | 84.4         |
> | SFT (GPT-4 + GPT-3.5)    | 52.7     | 78.6       | 33.1     | 46.3         |
> | SFT (GPT-3.5)            | 42.8     | 76.5       | 16.9     | 35.0         |
>
> - Second, note that `vicuna-v1.5-13b` and our `openchat-13b` uses the same base model (`llama-2-13b`), as well as ShareGPT datasets (both with GPT-4 + GPT-3.5 data) for finetuning, and `vicuna-v1.5-13b` uses even more data than ours (125k ShareGPT conversations vs 70k in ours), but our `openchat-13b` has achieved much better performance.
> - We have also compared with LLM models that are trained with SFT+RLFT as shown in the following table. Specifically, `llama-2-chat-13b` has the same base model as ours, but is tuned using SFT+RLFT with ~27k high-quality SFT data + ~2.9M preference, but its average score is still worse than our `openchat-13b`. This shows the strength of our proposed C-RLFT.
>
> Table R2: Comparing openchat with our C-RLFT to a series of fine-tuned LLMs with SFT + RLFT.
> | Models              | Base Models   | Method     | Dataset                                       | AlpacaEval | MT-bench | Vicuna-bench | Average  |
> | ------------------- | ------------- | ---------- | --------------------------------------------- | ---------- | -------- | ------------ | -------- |
> | **Larger than 13b** |               |            |                                               |            |          |              |          |
> | `gpt-4`             | -             | SFT + RLFT | -                                             | 95.3       | 82.5     | 90.0         | 89.3     |
> | `llama-2-chat-70b`  | `llama-2-70b` | SFT + RLFT | -                                             | 92.7       | 60.0     | 87.5         | 80.1     |
> | `claude`            | -             | SFT + RLFT | -                                             | 88.4       | 65.0     | 76.3         | 76.6     |
> | `gpt-3.5-turbo`     | -             | SFT + RLFT | -                                             | 86.1       | 50.0     | 50.0         | 62.0     |
> | **Equal to 13b**    |               |            |                                               |            |          |              |          |
> | `llama-2-chat-13b`  | `llama-2-13b` | SFT + RLFT | ~27k high-quality SFT data + ~2.9M preference | 81.1       | 55.3     | **86.9**     | 74.4     |
> | `openchat-13b`      | `llama-2-13b` | C-RLFT     | ~70k ShareGPT                                 | **89.5**   | **57.5** | 85.0         | **77.3** |
>
> - Lastly, our main focus of this paper is to provide a much simpler and extremely lightweight alternative solution to the existing costly RLFT methods for open-source LLM fine-tuning, rather than developing the most powerful LLM. Our proposed method can be very useful for developers with budget constraints or unable to collect lots of human preference data.

---

> ### Author Response · Authors · 2023-11-19
> **Response to Weakness 2 and Question 1**
>
> > **[W2]** It is not clear if the "low quality data" is even useful at all. (see my question below)
> >
> > **[Q1]** What is the performance if the llama model is only trained on GPT-4 data? I feel it should be another ablated model to demonstrate the value of "low quality data".
>
> **Response:**
>
> - The "lower quality data" is important in our C-RLFT method, as it needs to provide contrast signals to guide policy learning. As shown in the previous Table R1 in our response to **W1**, simply performing SFT on GPT-4 data cannot guarantee the best performance. Naively tuning with mixed GPT-4 and GPT-3.5 data will lead to even worse results. In contrast, C-RLFT greatly outperforms SFT on the same datasets. This is primarily because C-RLFT is performing RL rather than supervised learning on data. It uses the quality difference between two datasets as an implicit reward signal to guide policy learning, making the LLM model capable of discriminating good vs bad samples during learning, thus leading to improved performance. Without the lower-quality data, such contrast learning signals may no longer exist.
> - The performance `llama-2-13b` model finetuned on GPT-4 data using SFT is reported in the previous Table R1. We will also add the new results and related discussion in our revised paper.

---

### Official Review · Reviewer_QA7s · 2023-11-07

**Soundness:** 4 excellent
**Presentation:** 3 good
**Contribution:** 3 good
**Rating:** 6
**Confidence:** 4

**Summary:**

This paper presents a novel method to improve training quality of language models with mixed-quality data. Specifically, it consider the training data consists of good-quality expert data and low-quality data. Then it proposes conditioned RLFT that regards different data sources as coarse-grained reward labels and learns a class-conditioned policy to leverage complementary data quality information. Experimental evaluation shows that the proposed method achieves the highest average performance among all 13b open-source language models.

**Strengths:**

1. The proposed method is easy and efficient to train.
2. The method has many practical application scenarios where data quality can't be controlled very well especially for language model training.
3. The experimental evaluation confirms the effectiveness of the proposed method.

**Weaknesses:**

1. It requires the data quality to be divisible into two sets: good quality and average quality, which might potentially limit is applications.

**Questions:**

1. how sensitive is the results to the division of "good" data and "average" data? For example, how about if both sets of the data is of the same quality?
2. When training with Reinforcement Learning, how stable is the training process, since RL is usually different to train in most cases?

---

> ### Author Response · Authors · 2023-11-19
> **Response to Weakness 1**
>
> > **[W1]** It requires the data quality to be divisible into two sets: good quality and average quality, which might potentially limit is applications.
>
> **Response:**
>
> - Thanks for your comment. It is worth noting that our method actually has extremely relaxed requirements on datasets and is very friendly for applications. It only requires to have a small set of expert data and a larger set of medium-quality data, without the need for detailed quality evaluation for each sample nor costly human preference labels. In our experiments, we use the widely adopted ShareGPT dataset (~70k conversations), with only 6k expert conversations generated by GPT-4. This finetuning dataset essentially is the same one used to finetune `vicuna-v1.1-13b`, but as shown in our paper, our `openchat-13b` achieved significantly better performance.
> - We have also summarized the finetuning data sources of OpenChat and other popular open-source language models. It should be noted many existing models have more stringent requirements on the finetuning data, either data quantity or data quality. By contrast OpenChat enables to use the minimal amount of potentially expensive expert data as well as more easily obtainable medium-quality data to achieve improved LLM finetuning performance.
> - Lastly, we'd like to mention that the expert vs sub-optimal datasets division in C-RLFT can also be conducted according to the downstream tasks, e.g., safe vs not-so-safe data, preferred role vs. less preferred role in role-playing tasks, etc. Our proposed C-RLFT framework provides a very flexible and performant approach to address a wide range of applications.
>
> Table: The data source of the proposed OpenChat and other popular language models.
>
> | Model                     | Base Model    | Finetuning | Data                                             |
> | ------------------------- | ------------- | ---------- | ------------------------------------------------ |
> | `vicuna-v1.1-13b`         | `llama-13b`   | SFT        | ~70k ShareGPT (GPT-4 + GPT-3.5)                  |
> | `wizardlm-v1.0-13b`       | `llama-13b`   | SFT        | ~70k GPT-3.5                                     |
> | `ultralm-13b`             | `llama-13b`   | SFT        | ~1.5M UltraChat (GPT-3.5)                        |
> | `llama-2-chat-13b`        | `llama-2-13b` | SFT + RLFT | ~27k high-quality SFT data + ~2.9M preference    |
> | `vicuna-v1.5-13b`         | `llama-2-13b` | SFT        | ~125k ShareGPT (GPT-4 + GPT-3.5)                 |
> | `wizardlm-v1.2-13b`       | `llama-2-13b` | SFT        | ~250k Evol-Instruct + ShareGPT (GPT-4 + GPT-3.5) |
> | **`openchat-13b (ours)`** | `llama-2-13b` | C-RLFT     | ~70k ShareGPT (GPT-4 + GPT-3.5)                  |

---

> ### Author Response · Authors · 2023-11-19
> **Response to Question 1 and 2**
>
> > **[Q1]** How sensitive is the results to the division of "good" data and "average" data? For example, how about if both sets of the data is of the same quality?
>
> **Response:**
>
> - Thanks for the reviewer's constructive comment. It should be noted that if the "good" data and "average" data in C-RLFT are of the same quality, it actually reduces performing SFT on a single source of data. To fully address the reviewer's concern, we conducted additional experiments to compare C-RLFT on GPT-4 + GPT-3.5 against SFT methods tuned on GPT-4 and GPT-3.5, as well as their combination. The results are presented in the following table.
> - The results show that SFT tuned with only GPT-4 (\~6k) even performs better than SFT with both GPT-4 and GPT-3.5 (\~70k). This phenomenon confirms our statement in the paper that "it is not advisable to indiscriminatively feed all these mixed conversations to the base model, as the low-quality data are likely to negatively impact learning" in our Introduction. Furthermore, the performance of C-RLFT is still higher than SFT with GPT-4, proving the effectiveness of our proposed method. In our revised version, we will add this experiment to the ablation study.
>
> Table: Comparative performance of C-RLFT and SFT tuned on different data sources.
>
> | Type                     | Average  | AlpacaEval | MT-bench | Vicuna-bench |
> |--------------------------|----------|------------|----------|--------------|
> | C-RLFT (GPT-4 + GPT-3.5) | **77.3** | **89.5**   | **57.5** | **85.0**     |
> | SFT (GPT-4)              | 67.9     | 85.8       | 33.4     | 84.4         |
> | SFT (GPT-4 + GPT-3.5)    | 52.7     | 78.6       | 33.1     | 46.3         |
> | SFT (GPT-3.5)            | 42.8     | 76.5       | 16.9     | 35.0         |
>
> ---
>
> > **[Q2]** When training with Reinforcement Learning, how stable is the training process, since RL is usually different to train in most cases?
>
> **Response:**
>
> Thanks for your comment.
> - It should be noted that although C-RLFT optimizes the KL-regularized RL objective, its policy optimization can be equivalently transformed into a simple weighted regression objective. This actually allows us to solve an RL problem instead of using a single-stage, supervised learning scheme to learn the optimized policy.
> - Moreover, it should be noted that in our proposed C-RLFT, we **no longer** need to learn and optimize with a reward model, no need to involve the value networks as in PPO-based RLFT, and even not need to maintain the reference policy $\pi_0$ for policy constraint $D_{KL}(\pi_{\theta} \| \pi_0)$ as in most RLFT methods (also avoid related policy constraint weight parameter tuning). Everything we need is single-staged supervised learning, and thus is extremely lightweight and stable.

---

### Author Response · Authors · 2023-11-22
**Awaiting your valuable feedback before deadline**

Dear Reviewers,

Thanks for your insightful comments on our paper. We understand and appreciate the demands on your time, especially during this busy period. As we approach **the final day of the discussion stage**, we are reaching out to kindly request your feedback on our responses. Your insights are crucial to addressing any remaining concerns you may have regarding our submission.

We are particularly keen to engage in a constructive dialogue with you before the imminent deadline. Your feedback is not only valuable to us but also essential for the final evaluation of our work. We look forward to your response to our rebuttal.

Best regards,

The Authors

---

### Meta-Review · Area_Chair_bzFj · 2023-12-13

**Metareview:**

The paper presents a novel method designed to enhance language models using a combination of varied-quality data. The authors highlight the method's simplicity and efficiency. Extensive benchmarks show its strong performance among 13b open-source language models. No reviewer has strong concerns on this work, but the work can be improved in several aspects as highlighted by the reviewers. Specifically, concerns arise regarding the over-simplified estimation of data quality and dependence on assumptions, which may raise questions regarding generalizability. There are also discussions about the possible transfer of negative behaviors and adaptability challenges in complex data environments.

**Justification For Why Not Higher Score:**

Recommendations are not strong enough, and there are remaining concerns.

**Justification For Why Not Lower Score:**

All reviewers are on the positive side.

---

### Decision · Program_Chairs · 2024-01-16

Accept (poster)